# Residual apoptotic activity of a tumorigenic p53 mutant improves cancer therapy responses

Oleg Timofeev[1], Boris Klimovich[1], Jean Schneikert[1], Michael Wanzel[1,2], Evangelos Pavlakis[1], Julia Noll[1], Samet Mutlu[1], Sabrina Elmshäuser[1], Andrea Nist[3], Marco Mernberger[1], Boris Lamp[3], Ulrich Wenig[4], Alexander Brobeil[4], Stefan Gattenlöhner[4], Kernt Köhler[5] & Thorsten Stiewe[1,2,3,*] [iD]

## Abstract

Engineered p53 mutant mice are valuable tools for delineating p53 functions in tumor suppression and cancer therapy. Here, we have introduced the R178E mutation into the *Trp53* gene of mice to specifically ablate the cooperative nature of p53 DNA binding. *Trp53*[R178E] mice show no detectable target gene regulation and, at first sight, are largely indistinguishable from *Trp53*[−/−] mice. Surprisingly, stabilization of p53[R178E] in *Mdm2*[−/−] mice nevertheless triggers extensive apoptosis, indicative of residual wild-type activities. Although this apoptotic activity suffices to trigger lethality of *Trp53*[R178E];*Mdm2*[−/−] embryos, it proves insufficient for suppression of spontaneous and oncogene-driven tumorigenesis. *Trp53*[R178E] mice develop tumors indistinguishably from *Trp53*[−/−] mice and tumors retain and even stabilize the p53[R178E] protein, further attesting to the lack of significant tumor suppressor activity. However, *Trp53*[R178E] tumors exhibit remarkably better chemotherapy responses than *Trp53*[−/−] ones, resulting in enhanced eradication of p53-mutated tumor cells. Together, this provides genetic proof-of-principle evidence that a p53 mutant can be highly tumorigenic and yet retain apoptotic activity which provides a survival benefit in the context of cancer therapy.

**Keywords** apoptosis; Mdm2; mutant p53; p53; tumor suppression
**Subject Categories** Autophagy & Cell Death; Cancer; Molecular Biology of Disease
**The EMBO Journal (2019) 38: e102096**

See also: **JJ Manfredi** (October 2019)

## Introduction

The *TP53* gene, encoding the tumor-suppressive transcription factor p53, is mutated in about half of all human cancers. The presence of *TP53* mutations correlates in many cancer types with enhanced metastasis and aggressiveness, reduced responses to chemotherapeutic drugs, and, thus, a poor prognosis (Robles *et al*, 2016; Sabapathy & Lane, 2018). More than 85% of all amino acid positions were found to be mutated in cancer patients, generating a "rainbow" of > 2,000 distinct missense variants (Sabapathy & Lane, 2018). Mutations cluster in the central DNA binding domain (DBD), suggesting that tumorigenesis selects against p53's DNA binding function (Muller & Vousden, 2014; Stiewe & Haran, 2018). In support of this, mutant frequency was found to directly correlate with loss of transactivation function (Kato *et al*, 2003). However, *TP53* mutations show a remarkable preference for missense mutations, although DNA binding can be disrupted equally well by nonsense or frameshift mutations. Furthermore, missense mutants are unstable in normal unstressed cells, but become constitutively stabilized in tumors by Hsp90 which protects mutant p53 from degradation by Mdm2 and CHIP (Terzian *et al*, 2008; Alexandrova *et al*, 2015). The preferential selection of missense mutants together with their excessive stabilization therefore points at additional mechanisms that promote tumor development beyond a mere loss of DNA binding activity: Missense mutants exhibit dominant-negative effects on remaining wild-type p53 and display neomorphic properties that—like an oncogene—actively drive tumor development to a metastatic and drug-resistant state (Freed-Pastor & Prives, 2012; Muller & Vousden, 2014; Kim & Lozano, 2018; Stiewe & Haran, 2018). Missense mutations therefore enhance tumor development and progression in three ways: the loss of wild-type-like DNA binding activity (loss of function, LOF), dominant-negative effects on wild-type p53, and the gain of new tumor-promoting oncogenic properties (gain of function, GOF) (Stiewe & Haran, 2018).

p53 missense mutants are broadly classified as either "structural" or "DNA contact" mutants (Bullock & Fersht, 2001). Structural mutants destabilize the inherently low stability of the DBD resulting in its denaturation at body temperature and therefore likely affect also those non-transcriptional functions which are mediated by the DBD through, for instance, protein–protein

1 Institute of Molecular Oncology, Philipps-University, Marburg, Germany
2 German Center for Lung Research (DZL), Universities of Giessen and Marburg Lung Center, Marburg, Germany
3 Genomics Core Facility, Philipps University, Marburg, Germany
4 Institute of Pathology, Justus Liebig University, Giessen, Germany
5 Institute of Veterinary Pathology, Justus Liebig University, Giessen, Germany
*Corresponding author. Tel: +49 6421 28 26280; E-mail: thorsten.stiewe@uni-marburg.de

interactions. In contrast, DNA contact mutants affect single DNA-interacting residues and retain an intact native fold (Bullock & Fersht, 2001). We and others have previously described a new class of non-structural mutations affecting the DBD surface residues E180 and R181 which form a reciprocal salt bridge between two adjacent p53 subunits in the tetrameric DNA-bound complex (Klein *et al*, 2001; Kitayner *et al*, 2010). This salt bridge is essential for p53 to bind DNA in a cooperative manner so that mutations at these sites are referred to as cooperativity mutations (Dehner *et al*, 2005; Schlereth *et al*, 2010a). Despite being no mutational hot-spots, cooperativity mutations at residues E180 and R181 are estimated to account for 34,000 cancer cases per year (Leroy *et al*, 2014). Importantly, distinct cooperativity mutations reduce p53 DNA binding to different extents without changing the overall DBD structure determined by NMR spectroscopy (Dehner *et al*, 2005). Of all mutations of the double salt bridge, the R181E mutant disrupts formation of the intermolecular salt bridge most effectively (Schlereth *et al*, 2010a). Although R181E retains a native fold, it is entirely DNA binding deficient as assessed by electrophoretic mobility shift assays and genome-wide chromatin immunoprecipitation analysis (Dehner *et al*, 2005; Schlereth *et al*, 2010a, 2013). In R181E, the double salt bridge residues 180 and 181 are both glutamic acid (E), so that we refer to this mutant in short as "EE".

Mutations engineered into the mouse *Trp53* gene locus are valuable tools to delineate *in vivo* tumor suppressor functions in tumorigenesis and cancer therapy (Bieging *et al*, 2014; Mello & Attardi, 2018). Besides mutations derived from cancer patients, especially non-naturally occurring mutations of post-translational modification sites (Sluss *et al*, 2004; Slee *et al*, 2010; Li *et al*, 2012) or functional domains (Toledo *et al*, 2006; Brady *et al*, 2011; Hamard *et al*, 2013; Simeonova *et al*, 2013) have yielded substantial mechanistic insight into the pathways required for tumor suppression. To explore the relevance of DNA binding cooperativity for p53's anti-tumor activities, we therefore generated the "EE" mouse carrying the human R181E-equivalent R178E mutation at the endogenous *Trp53* gene locus. Cistrome and transcriptome analysis confirms the EE mutant as DNA binding deficient *in vivo*. Phenotype analysis demonstrates a knock-out-like appearance characterized by undetectable p53 target gene regulation and widespread, early-onset tumorigenesis, indicating that DNA binding cooperativity is essential for DNA binding and tumor suppression *in vivo*. Surprisingly, the EE mutation—different from the p53-knock-out—does not rescue the embryonic lethality of the *Mdm2* knock-out and triggers massive apoptotic cell death providing support for residual cytotoxic activities upon constitutive stabilization. An essential role of caspases, localization of EE to the mitochondria, and sensitization to mitochondrial outer membrane permeabilization point toward the intrinsic apoptosis pathway as the cause of cell death. Importantly, apoptosis was also triggered *in vitro* and *in vivo* by DNA-damaging chemotherapy of tumor cells expressing constitutively or pharmacologically stabilized EE. This translated into improved survival under chemotherapy. Similar results were obtained with the human R181L cooperativity mutant, which has been recurrently identified in cancer patients. Together, these findings highlight that mutant p53, in principle, can retain residual apoptotic activities that are insufficient to prevent tumorigenesis and not efficiently counter-selected during tumor evolution. Stabilization of such a p53 mutant in combination with chemotherapy is capable to trigger mutant p53-mediated cytotoxicity resulting in improved anti-cancer responses and increased survival.

# Results

## p53EE is deficient for DNA binding and target gene activation

We previously showed that the DNA binding cooperativity mutant p53$^{R181E}$ (EE) fails to bind p53 response elements *in vitro* and when exogenously expressed in p53-null cells (Schlereth *et al*, 2010a, 2013). To address how ablation of DNA binding cooperativity affects p53 functions *in vivo*, we generated a conditional knock-in mouse, carrying the R178E (EE) mutation in exon 5 of the endogenous mouse *Trp53* gene locus (Fig EV1A–D). DNA binding deficiency of the EE mutation in the context of the mouse p53 protein was confirmed by electrophoretic mobility shift assays using nuclear extracts of homozygous p53$^{EE/EE}$ mouse embryonic fibroblasts (MEFs) and a high-affinity, consensus-like p53 response element (Fig EV1E). Next, DNA binding was assessed genome-wide by sequencing chromatin immunoprecipitated with a p53 antibody from MEFs under untreated conditions and following p53 stabilization with the Mdm2 inhibitor Nutlin-3a (Nutlin) (ChIP-seq, Fig 1A). We identified a total of 468 p53-specific peaks in Nutlin-treated p53$^{+/+}$ MEFs (Figs 1A and EV1F). Validating the quality of the ChIP-seq, these peaks were strongly enriched for the p53 consensus motif at the peak center and significantly annotated with multiple Molecular Signatures Database (MSigDB) gene sets related to p53 function (Fig 1B and G). Only 3 peaks were identified in Nutlin-treated p53$^{EE/EE}$ MEFs that were, however, also called in p53$^{-/-}$ MEFs and therefore considered non-specific (Fig EV1F). Thus, the p53 binding pattern observed in p53$^{EE/EE}$ MEFs was indistinguishable from p53$^{-/-}$ MEFs, irrespective of Nutlin treatment, and therefore validated the p53EE mutant expressed from the endogenous *Trp53* gene locus to be DNA binding deficient in cells.

When global gene expression was profiled by RNA-seq, Nutlin exerted a significantly stronger effect on global gene expression in p53$^{+/+}$ versus either p53$^{EE/EE}$ or p53$^{-/-}$ MEFs, while Nutlin effects on p53$^{EE/EE}$ and p53$^{-/-}$ cells showed no significant difference (Fig EV1H). Furthermore, we observed in p53$^{+/+}$ but not in p53$^{EE/EE}$ or p53-null MEFs a strong Nutlin-inducible expression of a p53 signature including both *bona fide* p53 pathway genes (MSigDB Hallmarks_P53_Pathway) and non-canonical targets previously identified to be critical mediators of tumor suppression (Figs 1C and EV1I) (Brady *et al*, 2011). Gene set enrichment analysis (GSEA) showed a highly significant enrichment of a p53 target gene signature (MSigDB P53_Downstream_Pathway) in p53$^{+/+}$ cells compared to either p53$^{EE/EE}$ or p53$^{-/-}$ MEFs, but no enrichment in p53$^{EE/EE}$ versus p53$^{-/-}$ MEFs (Fig 1D). The same was observed for multiple other p53-related gene sets (Fig 1D). The lack of p53 target gene activation in p53$^{EE/EE}$ MEFs was confirmed also under conditions of DNA damage induced with doxorubicin (Fig 1F). Western blots revealed increased p53 expression in p53$^{EE/EE}$ versus p53$^{+/+}$ MEFs, which was further augmented by Nutlin or doxorubicin—yet in the absence of detectable expression of the p53 targets p21 and Mdm2 (Fig 1G and H). We conclude from these data that the murine p53EE mutant lacks detectable sequence-specific DNA binding and p53 target gene activation.

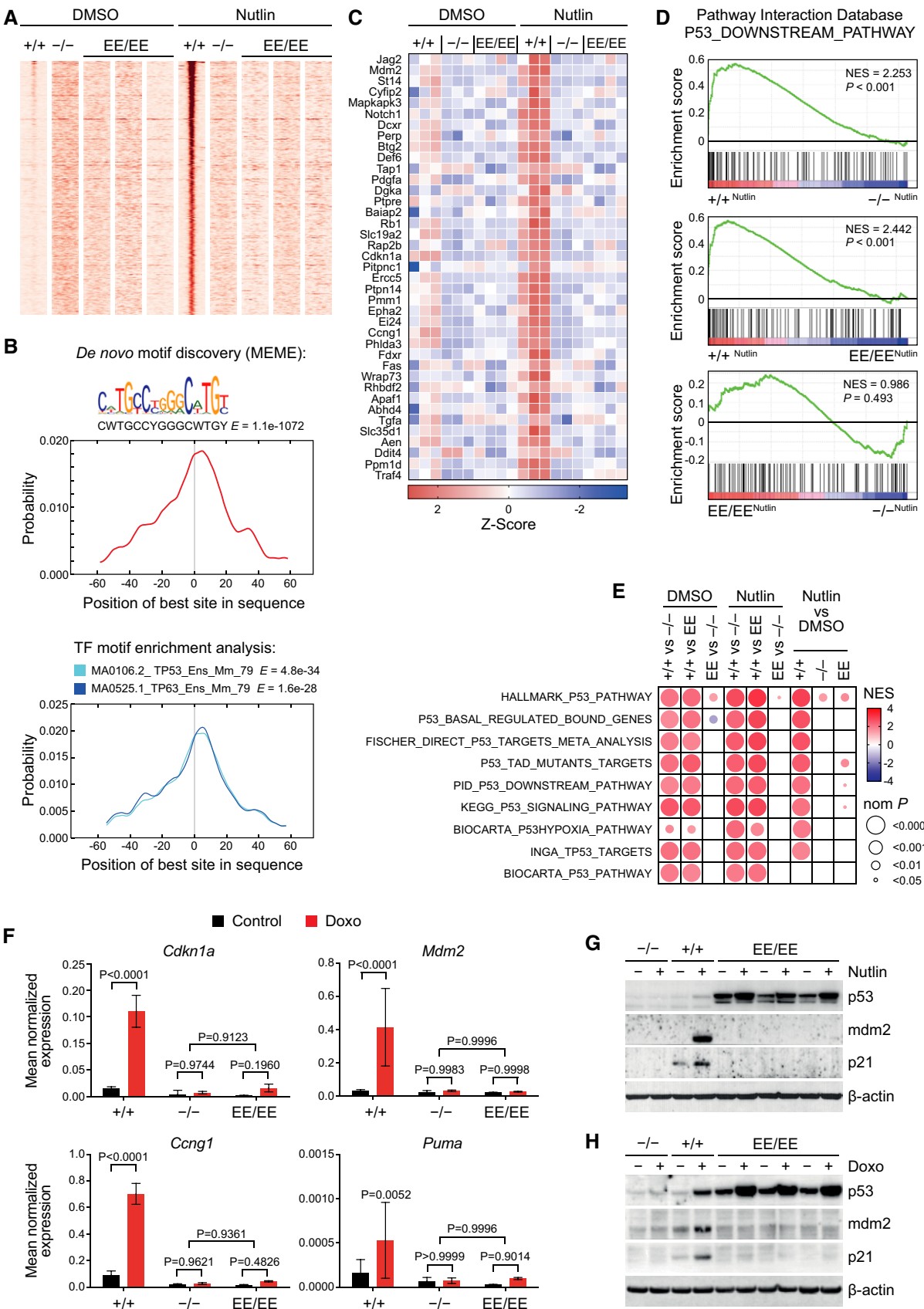

Figure 1.

◀

**Figure 1.  p53EE is deficient for DNA binding and target gene activation.**

A   p53 ChIP-seq in MEFs of the indicated genotype treated with or without 10 μM Nutlin-3a (Nutlin) for 16 h. Shown are 2 kb regions surrounding the summit of the 468 p53 binding peaks called in Nutlin-treated p53$^{+/+}$, but not p53$^{-/-}$ or p53$^{EE/EE}$ MEFs. For p53$^{EE/EE}$ MEFs, three independent replicates are shown.

B   *De novo* motif search using MEME-Chip was performed on the 468 p53 binding peaks (as in A). The best hit motif is reported with corresponding *E*-value and logo (upper part). Graphs depict a CentriMo enrichment analysis for the best MEME motif (middle) and for known transcription factor binding sites (bottom). The top two hits are shown with corresponding *E*-values.

C   RNA-seq was performed with MEFs of the indicated genotype untreated or treated with 10 μM Nutlin for 16 h. Shown are all Nutlin-regulated genes from the MSigDB gene set Hallmarks_P53_Pathway with a mean log2FC≥1 in p53$^{+/+}$ cells. Shown are the z-transformed RNA expression values (FPKM).

D, E   RNA-seq data were subjected to gene set enrichment analysis (GSEA). Shown are enrichment plots for the indicated set of p53 downstream genes in pairwise comparisons of Nutlin-treated MEFs with the indicated p53 genotypes. (E) Summary of GSEA results for p53-related gene sets. NES, normalized enrichment score; nom P, nominal P value.

F   Reverse transcription–quantitative PCR (RT–qPCR) analysis of p53 target genes in MEFs treated for 24 h with 1 μg/ml doxorubicin (Doxo). Shown are expression values normalized to β-actin as mean ± SD (*n* = 6). P values were calculated by 2-way ANOVA with Sidak's multiple comparisons test.

G, H   Western blots of protein lysates prepared from MEFs treated for 24 h with (G) 10 μM Nutlin or (H) 0.4 μg/ml Doxo.

## p53EE fails to induce apoptosis, cell cycle arrest, and senescence

In response to various types of stress, wild-type p53 elicits cell cycle arrest and senescence mediated by transcriptional activation of target genes, such as *Cdkn1a*/p21. Consistent with the inability of p53$^{EE}$ to induce target genes, p53$^{EE/EE}$ and p53-null MEFs comparably failed to undergo cell cycle arrest in response to doxorubicin-triggered DNA damage (Fig 2A) or to enter senescence upon enforced expression of oncogenic Ras (Fig EV2A) or *in vitro* passaging (Figs 2B and EV2B).

Besides cell cycle arrest, p53 is capable of inducing apoptotic cell death upon severe DNA damage. While immortalization with adenoviral E1A.12S strongly sensitized p53$^{+/+}$ MEFs to apoptosis, E1A-expressing p53$^{EE/EE}$ and p53-null MEFs remained refractory to apoptosis induction by genotoxic damage or Nutlin (Fig 2C). Likewise, p53$^{EE/EE}$ thymocytes were as resistant as p53-null cells to apoptosis triggered by ionizing radiation, despite retaining the ability to undergo p53-independent apoptosis thereby excluding a general failure of the apoptosis machinery (Fig 2D). The apoptosis defect corresponded with a deficiency in transactivating not only *Cdkn1a*/p21 but also the key pro-apoptotic target genes *Pmaip1*/Noxa and *Bbc3*/Puma (Fig 2E).

Of note, we have previously reported a similar but more selective apoptosis defect in p53RR mice carrying the E177R (RR) cooperativity mutation (Fig EV2C) (Timofeev *et al*, 2013). p53RR forms a p53WT-like salt bridge with p53EE which enables formation of stably DNA-bound and transcriptionally active p53EE/p53RR heterotetramers (Fig EV2C) (Dehner *et al*, 2005; Schlereth *et al*, 2010a, 2013). We therefore crossed p53$^{EE/EE}$ mice to p53$^{RR/RR}$ mice and obtained compound p53$^{EE/RR}$ animals that launched an apoptotic DNA damage response like p53$^{+/+}$ animals in thymocytes *ex vivo* (Fig EV2D) and upon whole-body irradiation *in vivo* (Fig EV2E). Rescue of the apoptosis deficiency of p53$^{EE/EE}$ mice by the equally apoptosis-defective p53RR mutant proves that the p53EE loss-of-function phenotype is directly linked to the inability to form the salt bridge responsible for cooperative DNA binding and in turn further excludes global DBD misfolding or secondary local structural alterations at the DNA binding surface as an underlying cause.

Like p53WT and the hot-spot mutant p53R172H (Terzian *et al*, 2008), p53EE was undetectable *in vivo* by immunostaining in all tissues analyzed, but rapidly stabilized in response to whole-body ionizing radiation (Fig EV2F). This suggests that the elevated p53EE protein level observed in MEF cultures (Fig 1G and H) reflects a stabilization in response to unphysiological culture stress. p53 stabilization upon whole-body irradiation triggered waves of cell cycle inhibition and apoptosis in intestinal crypts and other radiosensitive organs of p53$^{+/+}$ animals (Figs 2F–I and EV2G and H). None of these effects were recorded in p53$^{EE/EE}$ or p53-null mice (Figs 2F–I and EV2G and H), indicating a complete defect of p53EE regarding classical p53 effector functions *in vivo*.

## Constitutive p53EE stabilization triggers ROS-dependent senescence

When passaging p53$^{EE/EE}$ MEFs for longer time periods, we noted that —unlike p53$^{-/-}$ MEFs—the proliferation rate of p53$^{EE/EE}$ MEFs eventually declined and the cells started to express the senescence marker SA-β-galactosidase (Fig EV3A and B). This was accompanied by a progressive increase in p53EE protein levels, but without the increased expression in p53 target genes that was detectable in p53$^{+/+}$ MEFs (Fig EV3C and D). Spontaneous (Fig EV3C) or CRISPR-enforced (Fig EV3E) deletion of p53EE caused senescence bypass resulting in immortalization. Senescent p53$^{EE/EE}$ MEFs exhibited strongly elevated levels of mitochondrial ROS (Fig EV3F). Oxygen reduction from ambient to physiological levels promoted immortalization (Fig EV3G and H), implying ROS as the trigger of senescence in response to p53EE accumulation. However, there was no evidence for a p53WT-like inhibition of aerobic glycolysis (Warburg effect) and shift toward oxidative phosphorylation by p53EE which could explain an increased ROS production (Fig EV3I). Instead, we observed a somewhat reduced oxidative ATP production under basal conditions and significantly impaired spare respiratory capacity upon mitochondrial uncoupling, suggesting an inhibitory effect of p53EE on mitochondrial functions (Fig EV3I). In line, the mitochondrial DNA content of p53$^{EE/EE}$ MEFs was significantly decreased, especially at late passages (Fig EV3J).

A common regulator of ROS defense, oxidative phosphorylation, and mitochondrial biogenesis is the transcription factor Nrf2 (nuclear respiratory factor 2) (Dinkova-Kostova & Abramov, 2015). Basal and ROS-triggered expression of anti-oxidative Nrf2 target genes was diminished in p53$^{EE/EE}$ MEFs (Fig EV3K and L), strongly suggesting that p53EE inhibits Nrf2 activity similar as other p53 mutants (Walerych *et al*, 2016; Liu *et al*, 2017; Merkel *et al*, 2017). We therefore postulate that the Nrf2-inhibitory activity of p53EE sensitizes to ROS-induced senescence.

## p53EE is unable to rescue the lethality of Mdm2-null mice

In light of the senescence induced by progressive accumulation of p53 upon long-term cultivation (Fig EV3), we further explored

whether p53EE exerts anti-proliferative or cytotoxic activities when constitutively stabilized by *Mdm2* knock-out. Genetic ablation of *Mdm2* in mice causes early embryonic lethality due to massive apoptosis initiated as early as E3.5, whereas simultaneous disruption of the *Trp53* gene rescues the lethal phenotype (Jones *et al*, 1995; Montes de Oca Luna *et al*, 1995; Chavez-Reyes *et al*, 2003).

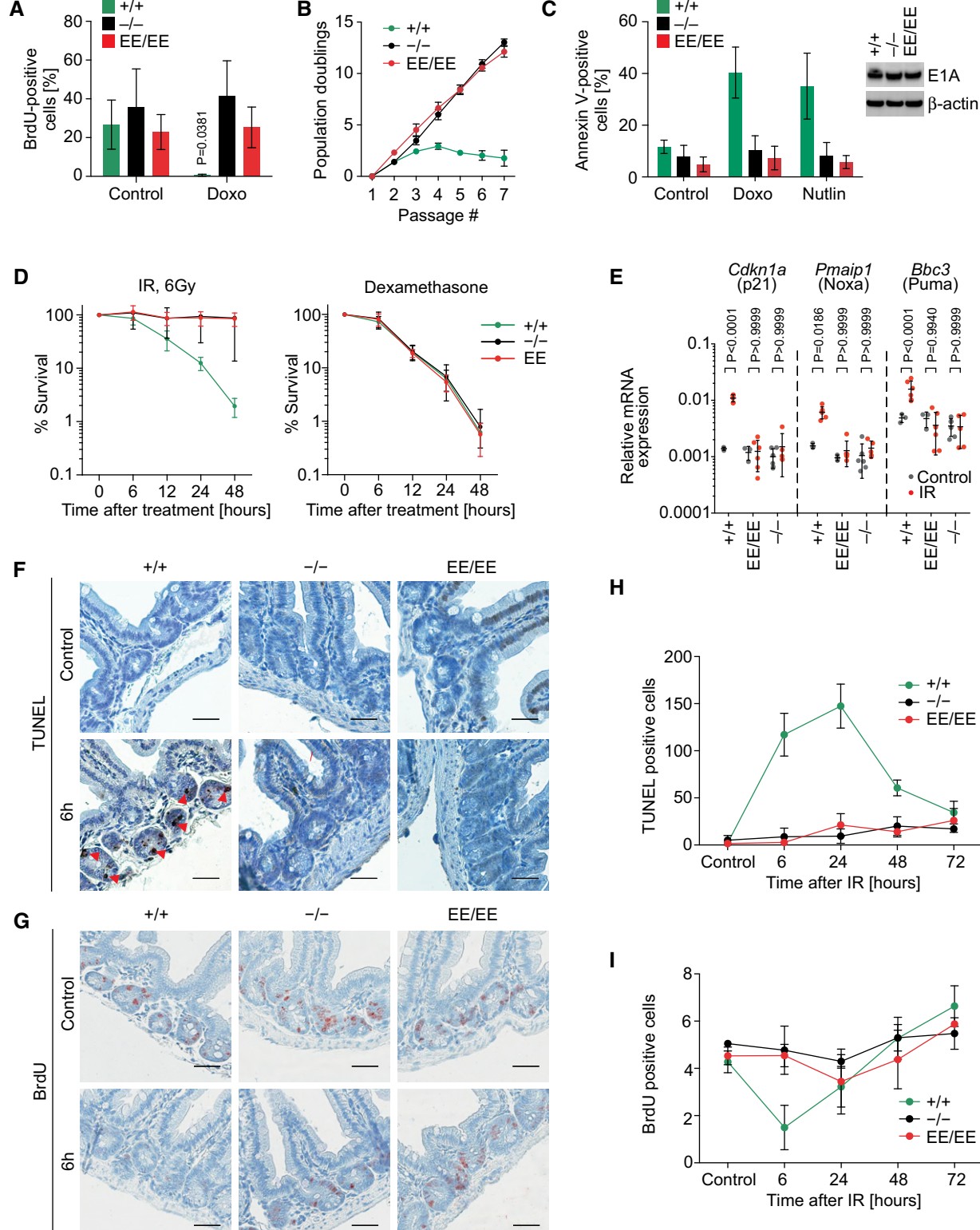

**Figure 2.**

**Figure 2. p53EE fails to induce apoptosis, cell cycle arrest, and senescence.**

A  Proliferation of primary MEFs. Cells were treated o/n with 0.2 µg/ml doxorubicin (Doxo) and pulse-labeled with 32 µM 5-bromo-2-deoxyuridine (BrdU), fixed and processed for flow cytometry analysis. $n = 4$.

B  Long-term proliferation assay with primary MEFs of indicated genotypes. +/+ and −/−: $n = 3$; EE/EE: $n = 6$.

C  MEFs were immortalized with the adenoviral oncogene E1A.12S (E1A MEF) and treated with 0.4 µg/ml Doxo for 17 h. Apoptosis (annexin V) was quantified by flow cytometry. +/+ and −/−: $n = 3$; EE/EE: $n = 6$. Western blots show expression of E1A and β-actin as loading control.

D  Primary thymocytes were irradiated ex vivo with 6 Gy or treated with 1 µM dexamethasone as a control for p53-independent apoptosis. Cell survival relative to untreated samples was analyzed using CellTiter-Glo assay (Promega). $n = 11$ for each time point and genotype.

E  mRNA expression analysis (RT–qPCR) of p53 target genes in thymocytes 6 h after 6 Gy irradiation. Shown are expression values normalized to β-actin.

F–I  Mice of indicated genotype were subjected to 6 Gy whole-body irradiation and pulse-labeled with 120 mg/kg BrdU 2 h before sacrifice at different time points. Small intestines were stained for (F) apoptosis (TUNEL) and (G) proliferation (BrdU); scale bars 50 µm. Red arrowheads highlight TUNEL-positive apoptotic cells. (H,I) Quantification for $n = 3$ mice/genotype (150 crypts/mouse).

Data information: All data are shown as mean ± SD. P values were calculated by 2-way ANOVA with Sidak's multiple comparisons test.

Embryonic lethality of $Mdm2$ knock-out animals is also rescued by p53 missense mutations such as $Trp53^{R246S}$ and $Trp53^{R172H}$ that mimic human hot-spot mutants $p53^{R249S}$ and $p53^{R175H}$, respectively (Lee et al, 2012). Moreover, the apoptosis-deficient mutant $p53^{R172P}$ also rescued $Mdm2^{-/-}$ embryos, but the newborn mice showed severe developmental defects and died soon due to ROS-dependent hematopoietic failure (Abbas et al, 2010). Surprisingly, we observed a clear deviation from the expected Mendelian distribution in intercrosses of double heterozygous $Trp53^{+/EE};Mdm2^{+/-}$ animals. Notably, no $Trp53^{EE/EE};Mdm2^{-/-}$ pups were born alive, indicating that those embryos died in utero (Fig 3A). In contrast, upon breeding $Trp53^{+/-};Mdm2^{+/-}$ mice as controls, viable double homozygous offspring was obtained as expected (Fig 3B). $Trp53^{EE/EE};Mdm2^{-/-}$ embryos displayed severe developmental defects starting at days E9.5-10.5 (Fig 3C and D), and no living $Trp53^{EE/EE};Mdm2^{-/-}$ embryos were recovered after day E12.5 (Fig 3D). Immunohistochemical analysis of tissue sections revealed strong accumulation of p53 protein in $Trp53^{EE/EE};Mdm2^{-/-}$ embryos compared to $Trp53^{EE/EE};Mdm2^{+/-}$ controls accompanied by high levels of apoptosis (Fig 3E). We conclude that $Trp53^{EE/EE};Mdm2^{-/-}$ embryos survive approximately 1 week longer than $Trp53^{+/+};Mdm2^{-/-}$ embryos which further underlines the functional defect of p53EE. Importantly, the failure of p53EE to completely rescue Mdm2-deficient embryos from lethality distinguishes p53EE from the p53 knock-out and reveals residual cytotoxic activities of p53EE with severe biological consequences in vivo.

However, the lethality of $Trp53^{EE/EE};Mdm2^{-/-}$ embryos contrasted with the lack of any detectable cytotoxic activity of the Mdm2 inhibitor Nutlin in p53$^{EE/EE}$ MEFs (Fig 2C). Of note, Nutlin specifically disrupts the interaction between p53 and Mdm2 leading to p53 stabilization, but does not inhibit many other Mdm2 functions, such as its p53-independent metabolic role in ROS detoxification (Riscal et al, 2016). As the p53EE-mediated senescence in late passage cultures was linked to both p53 accumulation and ROS (Fig EV3), we hypothesize that the lethality of $Trp53^{EE/EE};Mdm2^{-/-}$ embryos also results from the combination of p53EE stabilization and increased ROS, which are both consequences of $Mdm2$ ablation.

## Pharmacological inhibition of Mdm2 unleashes cytotoxic activities of p53EE

This prompted us to investigate whether accumulation of p53EE caused by loss of Mdm2 sensitizes cells to doxorubicin whose cytotoxicity involves ROS- and DNA damage-dependent mechanisms

(Trachootham et al, 2009; Huang et al, 2011). Because of the embryonic lethality, we could not establish $Trp53^{EE/EE};Mdm2^{-/-}$ MEFs and used $Trp53^{-/-};Mdm2^{-/-}$ (DKO) MEFs ectopically expressing p53EE from a tetracycline-activated promoter instead (DKO-tetEE). As expected, in the absence of tetracycline the DKO-tetEE MEFs were as resistant to doxorubicin as the parental DKO cells but showed significantly elevated levels of apoptosis in response to doxorubicin treatment following induction of p53EE expression (Fig 4A). In compliance with the DNA binding deficiency of p53EE, this was not accompanied by transcriptional activation of key pro-apoptotic p53 target genes (Fig 4B).

Next, we tested whether pharmacological inhibition of Mdm2 with Nutlin has a similar effect on E1A-MEFs with endogenous expression of p53EE. While Nutlin or doxorubicin alone had no or minimal effects on p53$^{EE/EE}$ MEFs, combined treatment of p53$^{EE/EE}$ cells with Nutlin and doxorubicin caused significant, p53-dependent reduction in proliferation and survival (Fig 4C and D). Similar cytotoxic activity was also observed when doxorubicin was combined with other Mdm2 inhibitors (Fig 4E), excluding toxic off-target effects of Nutlin as an explanation. Cell death involved cleavage of Parp and caspase-3, but upregulation of p53 target genes was not detectable (Fig 4F and G).

The Hsp90 inhibitor ganetespib, which targets mutant p53 for proteasomal degradation, efficiently degraded p53EE in E1A-MEFs, highlighting a role for Hsp90 in p53EE stability (Fig 4H). p53EE degradation itself was not associated with apoptosis, indicating that E1A-MEFs are not dependent on p53EE. In line with the role of Mdm2 for ganetespib-induced mutant p53 degradation (Li et al, 2011), p53EE degradation by ganetespib was prevented when administered simultaneously with Nutlin (Fig 4H). Pre-treatment with ganetespib, however, led to p53EE degradation and efficiently counteracted the apoptosis induced by sequential treatment with Nutlin plus doxorubicin, thereby validating stabilized p53EE as the mediator (Fig 4H).

As wild-type p53 exerts direct, non-transcriptional, pro-apoptotic effects at the mitochondria (Mihara et al, 2003; Chipuk et al, 2004; Leu et al, 2004; Le Pen et al, 2016), we tested for mitochondrial localization of p53EE (Fig 4I). Remarkably, mitochondrial fractions of untreated p53$^{EE/EE}$ MEFs contained p53 at levels similarly high as p53$^{+/+}$ cells following doxorubicin treatment. The amount of mitochondrial p53EE increased even further under doxorubicin treatment. The increased amount of p53EE at the mitochondria could therefore provide a plausible explanation for the sensitivity of p53$^{EE/EE}$ MEFs to apoptotic stimuli even in the absence of p53 target gene activation.

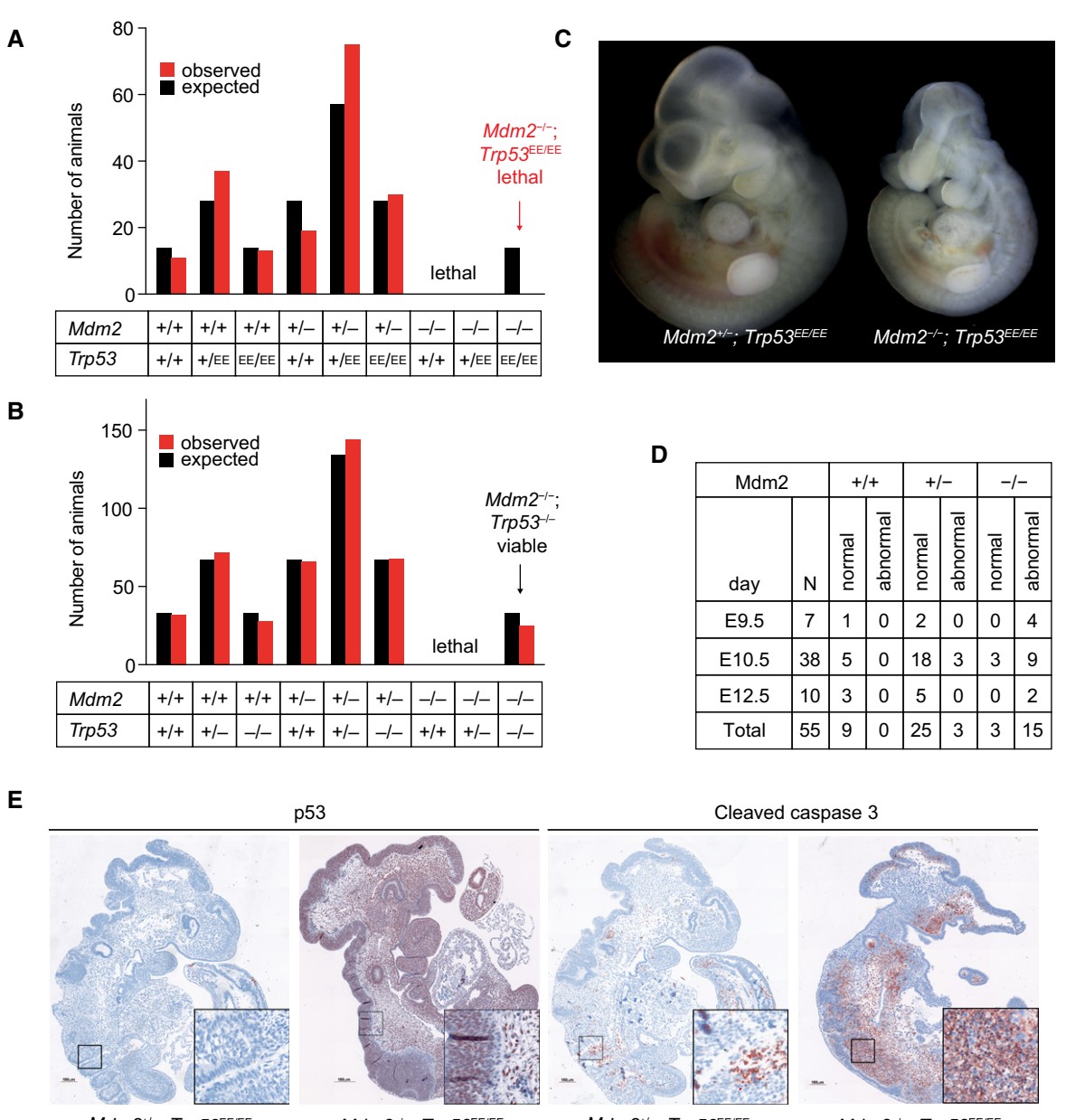

**Figure 3. Lethality of p53EE embryos in the absence of Mdm2.**

A   Observed and expected genotype distribution of newborn offspring from mating $Mdm2^{+/-} Trp53^{EE/+}$ mice (total number of pups $n = 185$; contingency test $P = 0.0029$).
B   Observed and expected genotype distribution of newborn offspring from mating $Mdm2^{+/-} Trp53^{+/-}$ mice (total number of pups $n = 435$; contingency test $P = 0.912$).
C   Representative picture of an E9.5 $Mdm2^{-/-}Trp53^{EE/EE}$ embryo (right) displays characteristic phenotypic abnormalities compared to an $Mdm2^{+/-};Trp53^{EE/EE}$ embryo (left).
D   Phenotype analysis of $Trp53^{EE/EE}$ embryos with different $Mdm2$ genotypes reveals developmental defects in $Mdm2^{-/-}Trp53^{EE/EE}$ embryos starting from E9.5.
E   IHC for p53 and cleaved caspase-3 shows strong accumulation of p53EE protein and massive apoptosis in $Mdm2^{-/-}Trp53^{EE/EE}$ compared to $Mdm2^{+/-};Trp53^{EE/EE}$ control embryos.

## Residual apoptotic activities of mutant p53 in human cancer cells

To test whether p53EE is also capable of inducing apoptosis in human cancer cells, we generated a human p53-null H1299 lung cancer cell line with stable Tet-inducible expression of the human p53$^{R181E}$ (p53EE) mutant. Overexpression of p53EE rendered H1299 cells, which express only barely detectable levels of Mdm2, sensitive to doxorubicin, and addition of Nutlin further augmented this effect (Fig 5A). Cell death was inhibited by Q-VD-OPh, a blocker of caspase-dependent apoptosis, but not by the ferroptosis inhibitor ferrostatin as a control (Fig 5B). Annexin V staining revealed a p53-dependent increase in apoptotic cells under combined doxorubicin and Nutlin treatment, confirming the observed cell death as apoptotic (Fig 5C). p53EE-mediated apoptosis in H1299 cells occurred in

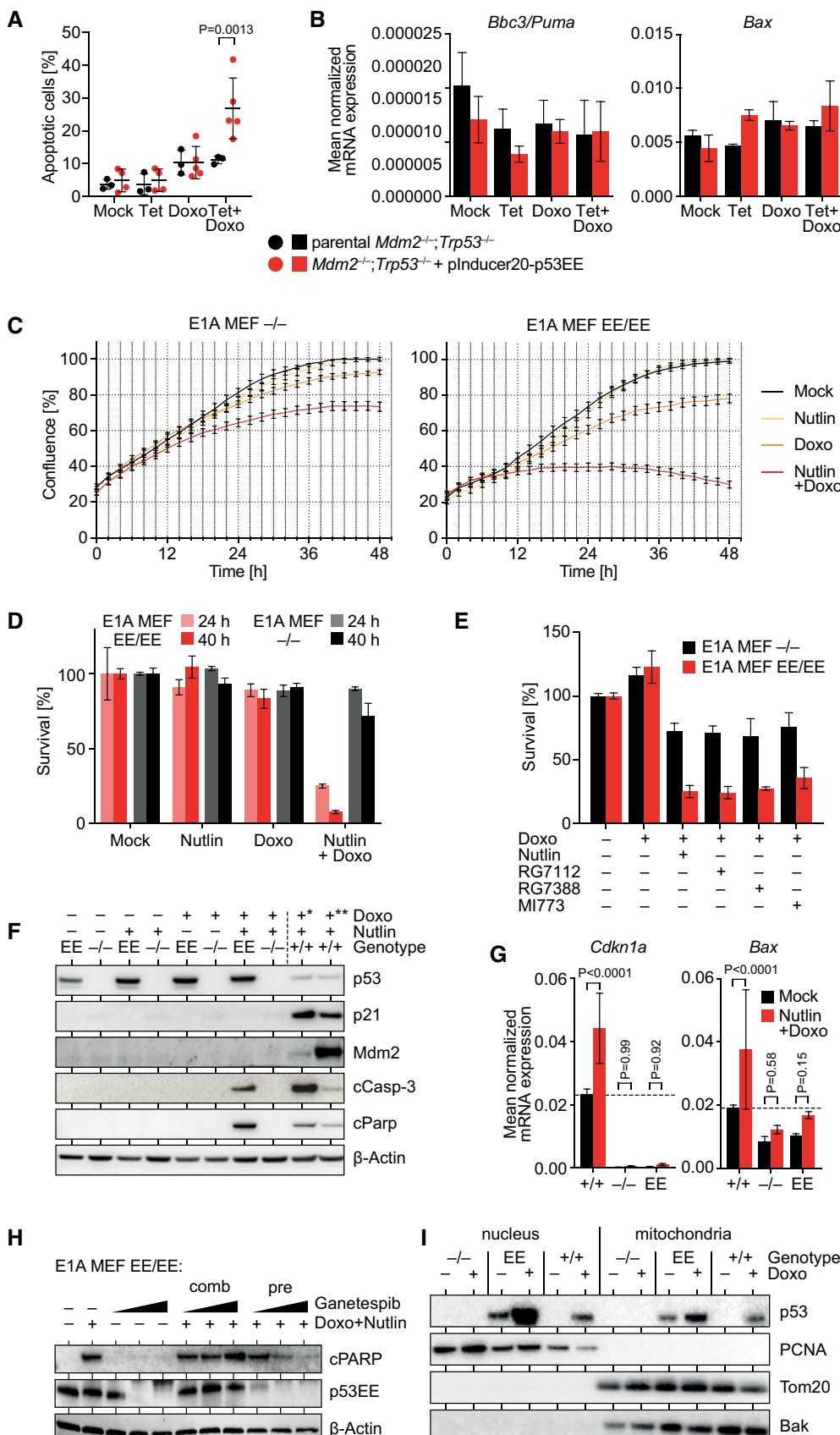

Figure 4.

**Figure 4. Pharmacological inhibition of Mdm2 unleashes cytotoxic activities of p53EE.**

A   *Mdm2*$^{-/-}$*Trp53*$^{-/-}$ (double knock-out, DKO) MEFs were transduced with pInducer20-p53EE lentivirus to enable tetracycline (Tet)-inducible expression of mouse p53EE. Induction of p53EE sensitized cells to apoptosis induced by 24-h treatment with 1 μg/ml doxorubicin (Doxo) as detected by flow cytometry for annexin V.

B   mRNA expression of p53 target genes was measured in cells from (A) relative to β-actin by RT–qPCR following treatment with Tet ± 1 μg/ml Doxo. Expression of the pro-apoptotic p53 target genes *Bbc3* (Puma) and *Bax* is not significantly induced. *n* = 6.

C   Proliferation of MEFs with indicated genotypes was analyzed by live-cell imaging in the presence of 10 μM Nutlin-3a (Nutlin) and/or 0.05 μg/ml doxorubicin (Doxo). Shown is the median confluence ± SEM (*n* = 12).

D, E   Cell viability assays for MEFs with indicated genotypes treated with Mdm2 inhibitors ± 0.4 μg/ml Doxo. *n* = 3.

F   Western blot of E1A-MEFs with indicated genotypes treated with 10 μM Nutlin ± 0.4 μg/ml Doxo for 18 h reveals induction of apoptosis (cCasp-3, cleaved caspase-3; cParp, cleaved Parp) in double-treated p53$^{EE/EE}$ MEFs in the absence of p53 target gene (p21, Mdm2) activation. *10 h 0.4 μg/ml Doxo; **10 h 0.2 μg/ml Doxo.

G   mRNA expression of p53 target genes was measured in E1A MEFs of indicated genotype relative to β-actin by RT–qPCR following combined treatment with 10 μM Nutlin and 1 μg/ml Doxo. *n* = 12.

H   Western blot of p53$^{EE/EE}$ E1A-MEFs with indicated genotypes treated for 18 h with 10 μM Nutlin, 0.4 μg/ml Doxo, and 30-100-500 nM ganetespib as indicated. Comb, combined treatment with 3 drugs for 18 h. Pre, pre-treatment with ganetespib for 24 h followed by combined treatment with Nutlin and doxorubicin for 18 h.

I   Western blot of cellular fractions from unstressed and Doxo-treated MEFs of indicated genotype identifies p53EE protein in the mitochondrial fraction. PCNA and Tom20/Bak are shown as nuclear and mitochondrial marker proteins.

Data information: All data are shown as mean ± SD unless indicated otherwise. Significance was tested between the two cell lines or treatments. *P* values were calculated by 2-way ANOVA with Sidak's multiple comparisons test.

the absence of detectable p53 target gene activation (Fig 5D). Proximity ligation experiments demonstrated a specific co-localization of p53EE with the outer mitochondrial membrane protein Tom20 in untreated and, even more, in doxorubicin-treated cells (Fig 5E and F). p53EE co-localization was also observed with the Bcl-2 family members Bcl-xL, Bcl-2, and Bak (Fig 5F), which were previously shown to specifically interact with wild-type but not mutant p53 (Mihara *et al*, 2003; Leu *et al*, 2004; Pietsch *et al*, 2008). Mitochondrial activities of wild-type p53 "prime" mitochondria to mitochondrial depolarization by BH3 peptides (Montero *et al*, 2015; Le Pen *et al*, 2016). In support of a direct mitochondrial apoptotic priming activity, p53EE expression sensitized H1299 cells to mitochondrial depolarization by BID BH3 peptides (Fig 5G).

The apoptotic activity of p53EE raised the question of whether p53 mutants in cancer patients can also retain apoptotic activity. Although p53EE (R181E) is not found in cancer patients, various other cooperativity mutations have been recurrently identified as somatic cancer mutations and account for an estimated 34,000 new cancer cases each year (Leroy *et al*, 2014). R181 mutations were also reported as germline alterations associated with a family history of cancer, identifying them as *bona fide* driver mutations (Frebourg *et al*, 1992; Leroy *et al*, 2014). In particular, R181L was among the first p53 cancer mutants reported to be specifically deficient for binding and activating pro-apoptotic target genes while retaining regulation of other genes such as *CDKN1A* and *MDM2* (Ludwig *et al*, 1996; Schlereth *et al*, 2010a). Consistently, R181L failed to induce apoptosis when overexpressed irrespective of Nutlin treatment (Fig 5H). However, R181L—like p53EE—effectively triggered apoptosis when combined with doxorubicin and Nutlin (Fig 5H). As with p53EE, apoptosis occurred without activation of the key pro-apoptotic p53 target gene *Puma* (Fig 5I). In contrast, the R175P mutant, which also lacks the ability to transactivate pro-apoptotic target genes but belongs to the class of structural mutations (Rowan *et al*, 1996; Liu *et al*, 2004), failed to trigger apoptosis in the presence of doxorubicin and Nutlin (Fig 5H). We conclude that, in principle, also cancer mutants can trigger apoptosis when stimulated sufficiently with Nutlin and doxorubicin in a manner depending on the identity of the individual mutant.

## p53EE fails to suppress tumor development

To investigate the role of DNA binding cooperativity for tumor suppression, we aged cohorts of EE mutant mice. Surprisingly, despite evidence for residual apoptotic activity of p53EE (Figs 3–5), homozygous p53$^{EE/EE}$ and hemizygous p53$^{EE/-}$ mice developed tumors very rapidly resulting in a short median survival of 150 and 128 days, respectively (Fig 6A). This was not significantly different from p53$^{-/-}$ animals. Heterozygous p53$^{EE/+}$ showed an intermediate median survival of 519 days that was not significantly altered as compared to p53$^{+/-}$ mice (Fig 6A). The lack of a difference in survival between p53$^{EE/+}$ and p53$^{+/-}$ mice, despite evidence for a dominant-negative activity of p53EE in overexpression studies (Schlereth *et al*, 2010a), is reminiscent of mice heterozygous for the hotspot mutants R172H and R270H and consistent with the hypothesis that dominant-negative effects might require prior mutant p53 stabilization (Lang *et al*, 2004; Olive *et al*, 2004).

As previously reported for p53$^{-/-}$ animals, the tumor spectra of p53$^{EE/EE}$ and hemizygous p53$^{EE/-}$ were dominated by thymic lymphoma (Fig 6B; Appendix Table S1). Additional tumor types were B-cell lymphomas, sarcomas, and testicular tumors (Appendix Table S1). The tumors of p53$^{EE/+}$ were similar to those of p53$^{+/-}$ mice and mostly non-thymic lymphomas, sarcomas, and several types of carcinoma (Fig 6B, Appendix Table S2).

Unlike what has been reported for the p53 hot-spot mutants R172H or R270H, we did not observe an increase in metastatic tumors in p53$^{EE/EE}$ and p53$^{EE/-}$ compared to p53$^{-/-}$ and p53$^{+/-}$ mice (Appendix Table S1), suggesting a lack of GOF properties of the EE mutant. For hot-spot mutants, constitutive stabilization was identified as a prerequisite for GOF effects (Terzian *et al*, 2008). Similar as described for tumors in R172H-mutant mice (Terzian *et al*, 2008), we noted varying levels of p53EE expression in spontaneous tumors arising in p53$^{EE/EE}$ or p53$^{EE/-}$ mice, with a high fraction (67%) of p53$^{EE/EE}$ tumors exhibiting p53EE stabilization comparable to p53-mutated human cancer samples (Figs 6C and EV4 and EV5). This indicates that the absence of a pro-metastatic GOF cannot be explained by a lack of constitutive p53EE stabilization and suggests that p53EE, similar to p53 mutants like R246S or

G245S (Lee *et al*, 2012; Hanel *et al*, 2013), lacks the pro-metastatic GOF activity described for R172H and R270H (Olive *et al*, 2004).

To explore whether p53EE can counteract oncogene-induced tumorigenesis in a genetically more defined setting, we crossed p53EE animals with Eμ-Myc mice which serve as a

well-characterized model of Burkitt-like B-cell lymphoma (Adams *et al*, 1985). Loss of p53 is known to strongly accelerate lymphoma development in this model (Schmitt *et al*, 1999). We observed that Eμ-Myc;p53$^{EE/+}$ mice survived markedly shorter than Eμ-Myc; p53$^{+/+}$ but not significantly longer than Eμ-Myc;p53$^{+/-}$ mice

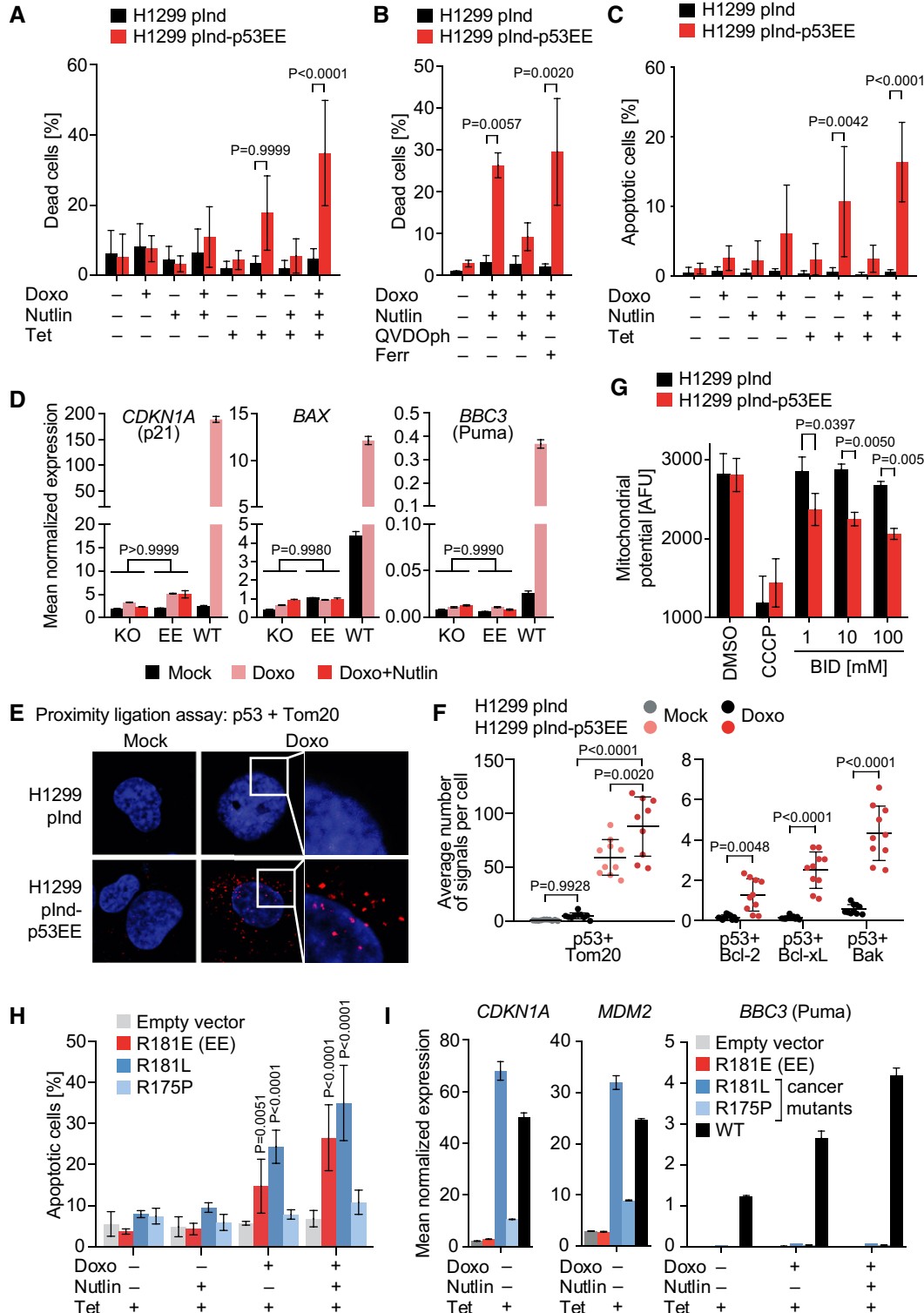

Figure 5.

**Figure 5.  Residual apoptotic activities of mutant p53 in human cancer cells.**

A, B  Cell death measured by flow cytometry using propidium iodide exclusion in human lung adenocarcinoma H1299 cells with Tet-inducible expression of human p53EE. Cells with overexpression of human p53EE mutant display a trend of increased cell death after treatment with 0.5 μg/ml Doxo. This effect is enhanced by addition of Nutlin and blocked by the pan-caspase inhibitor QVDOph, but not the ferroptosis inhibitor ferrostatin-1 (Ferr) as control. (A) $n = 3$; (B) $n = 2$.

C  Apoptosis analysis using annexin V staining in cells from (A). $n = 3$.

D  mRNA expression analysis (RT–qPCR) of p53 target genes following treatment with 0.5 μg/ml Doxo ± 10 μM Nutlin. KO, Tet-induced H1299-pInd cells; EE, Tet-induced H1299-pInd-p53EE cells; WT, H460 cells. mRNA expression was normalized to β-actin; $n = 3$.

E  Proximity ligation assay for p53 with Tom20 in indicated untreated (Mock) or Doxo-treated (0.1 μg/ml, 24 h) H1299 cells following Tet induction. Shown are representative cells with PLA signals (red) counterstained with DAPI (blue).

F  Quantification of proximity ligation assays for p53EE with Tom20, Bcl-2, Bcl-xL, and Bak in H1299 cells. Doxo treatment (0.1 μg/ml, 24 h) as indicated. Plotted is the average number of PLA signals per cell for 10 fields of view.

G  Mitochondrial membrane potential of control and p53EE-expressing H1299 cells in the absence and presence of increasing concentrations of BID BH3 peptide. Treatment with the mitochondrial depolarizer CCCP is shown as a positive control; $n = 3$.

H  Apoptosis measured by flow cytometry using annexin V in H1299 cells with Tet-inducible expression of the indicated human p53 mutants; $n = 3$.

I  mRNA expression analysis (RT–qPCR) of p53 target genes normalized to β-actin; $n = 3$.

Data information: All data are shown as mean ± SD. Significance was tested by 2-way ANOVA with Sidak's multiple comparisons test.
Source data are available online for this figure.

(Fig 6D). Likewise, in a model of acute myeloid leukemia induced by co-expression of the AML1/ETO9a (AE9) fusion oncoprotein and oncogenic Nras$^{G12D}$ (Nras) (Zuber *et al*, 2009), we observed fast malignant transformation of p53$^{EE/EE}$ hematopoietic stem cells (HSCs) and disease progression. Recipient mice transplanted with AE9+Nras-transduced p53$^{EE/EE}$ or p53$^{-/-}$ HSCs succumbed to AML with indistinguishably short latency and significantly earlier than mice transplanted with AE9+Nras-transduced p53$^{+/+}$ HSCs (Fig 6E). Notably, distinct from heterogenous p53EE stabilization in spontaneous tumors (Figs 6C and EV5), p53EE was highly accumulated in all lymphomas from Eμ-Myc;p53$^{EE/+}$ mice and in all AE9+Nras;p53$^{EE/EE}$ AML samples (Fig 6F and G). As stabilization of mutant p53 involves protection from Mdm2-mediated degradation (Terzian *et al*, 2008), the uniform p53EE accumulation might be explained by the exceptionally strong oncogenic signaling through enforced expression of Myc and mutant Nras which inhibits Mdm2 via p19Arf (Zindy *et al*, 1998).

Taken together, these data show that p53EE is inadequate to counteract spontaneous and oncogene-induced tumorigenesis in mice, proving DNA binding cooperativity to be absolutely essential for tumor suppression by p53 even in the presence of residual apoptotic functions.

**p53EE confers survival benefit under chemotherapy**

The lack of detectable tumor suppressor activity implies that the residual apoptotic activities are not effectively counter-selected during tumorigenesis and can be retained by cancer mutants—the R181L cooperativity mutant being one example (Fig 5H). In fact, p53EE was even constitutively stabilized in various tumor types arising in p53$^{EE/EE}$ mice (Fig 6), indicating escape from Mdm2-mediated degradation. We therefore explored whether this accumulation of p53EE suffices to sensitize tumor cells to cytotoxic stress and provide a therapeutic window for tumor treatment. First, we investigated whether p53EE influences the response of Eμ-Myc lymphoma cells to mafosfamide (MAF), a cyclophosphamide (CTX) analogue active *in vitro*. We established lymphoma cell lines from Eμ-Myc mice with different p53 genotypes (p53$^{+/+}$, p53$^{+/-}$, and p53$^{+/EE}$). Lymphomas from p53$^{+/EE}$ mice (and p53$^{+/-}$ mice) showed the expected loss of the wild-type allele and strongly expressed p53EE as described above (Fig 6F). In comparison with

the rapid and strong response of p53$^{+/+}$ lymphoma cells to MAF treatment, induction of apoptosis started in p53EE cells only after 6 h and gradually increased up to significantly elevated levels of 60–70% at 24 h, whereas p53-null lymphoma cells showed no response (Fig 7A). To test whether the increased sensitivity of p53EE lymphoma cells is directly p53EE-mediated, the lymphoma cells were transduced with a Tet-inducible, red fluorescence protein (RFP)-coupled shRNA to knockdown p53EE expression. RFP-positive, i.e., p53 shRNA-expressing, lymphoma cells became significantly enriched under MAF treatment in a dose-dependent manner, identifying the enhanced cytotoxic response of p53EE lymphoma cells as directly p53EE-mediated (Fig 7B). In support of a non-transcriptional mechanism, the cytotoxic response was not preceded by an upregulation of p53 target genes (Fig 7C).

Next, we transplanted primary lymphomas into syngeneic recipients. After lymphomas became palpable in the peripheral lymph nodes, each cohort was divided into two groups—one was subjected to a single injection of 300 mg/kg CTX, and the second was left untreated (Fig 7D). Independent of p53 genotype, the disease progressed rapidly with similar kinetics in untreated control mice, whereas all treated animals responded well to CTX therapy and went into clinical remission. In line with previous studies (Schmitt *et al*, 1999, 2002b), p53$^{-/-}$ lymphomas rapidly relapsed in all mice, which resulted in a very modest median survival benefit of 19 days. In contrast, all mice transplanted with p53$^{+/+}$ lymphomas remained in complete remission during the whole period of observation (90–180 days after treatment). Importantly, the chemotherapy was also effective for treatment of p53EE lymphoma, provided a median survival benefit of 30 days, and—even more compelling— yielded a complete tumor-free remission in 36% of animals (Fig 7D and E). To monitor the therapy response quantitatively, we measured residual disease in the spleen with a sensitive qPCR copy number assay for the Eμ-Myc transgene of moribund lymphoma mice and non-transgenic littermates as positive and negative controls, respectively (Fig 7F). Tumor load in mice with p53$^{+/+}$ lymphoma decreased by > 3 orders of magnitude within 24 h after treatment and dropped below the detection limit by 7 days, consistent with the uniform long-term clinical remission in this cohort. In line with the delayed apoptotic response of p53EE lymphoma cells *in vitro* (Fig 7A), we observed only a slight, but significant, decline in tumor load at 24 h in p53EE compared to p53$^{-/-}$ lymphoma

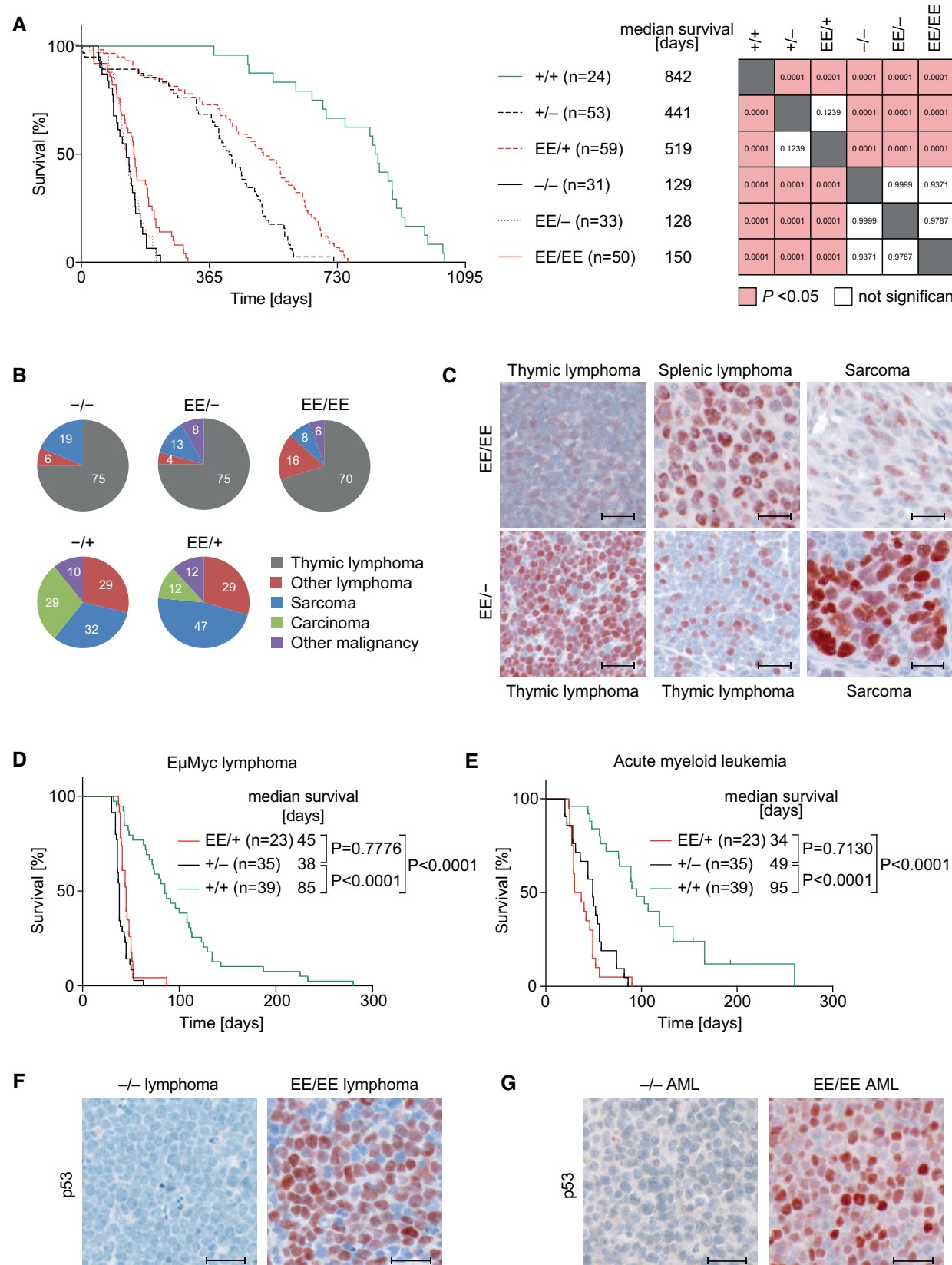

Figure 6.

Figure 6. p53EE fails to suppress tumor development.

A   Kaplan–Meier survival plots for mice with indicated genotypes, number of mice, and median survival. *Right panel*, analysis of survival differences using the log-rank (Mantel–Cox) test.
B   Spectra of malignant tumors found in mice with indicated genotypes (p53$^{-/-}$ $n$ = 16; p53$^{EE/-}$ $n$ = 24; p53$^{EE/EE}$ $n$ = 37; p53$^{+/-}$ $n$ = 28; p53$^{EE/+}$ $n$ = 17).
C   p53 immunohistochemistry (IHC) in representative spontaneous tumors; scale bars 25 μm.
D   Kaplan–Meier survival plots for Eμ-Myc transgenic mice with different p53 genotype.
E   Kaplan–Meier survival plots for wild-type recipient mice transplanted with AML1/ETO9a + Nras$^{G12D}$-transduced fetal liver cells (5 donors/genotype).
F, G   p53 IHC for representative samples from (D) and (E) illustrates constitutive stabilization of p53EE in cancer cells; scale bars 25 μm.

Data information: Survival differences were analyzed using the log-rank (Mantel–Cox) test. Multiple survival curves were compared by ordinary ANOVA with Tukey's multiple comparisons test.

mice. By 7 days, however, tumor cells had become undetectable in 3 of 5 p53EE lymphoma mice, indicating molecular remission, whereas p53$^{-/-}$ lymphoma cells remained detectable in all animals of the p53$^{-/-}$ group.

To validate in a second independent model that p53EE enhances the chemotherapy response *in vivo*, we transplanted syngeneic recipient mice with AE9/Nras-driven acute myeloid leukemia cells with different p53 genotypes and monitored disease progression by bioluminescence imaging (Fig 7G). p53$^{+/+}$ leukemia cells responded well to a standard chemotherapy protocol with cytarabine (AraC) combined with doxorubicin, which yielded a significant survival benefit (Fig 7G and H). p53$^{-/-}$ leukemia rapidly killed the transplanted animals irrespective of therapy. In striking contrast, chemotherapy controlled progression of p53$^{EE/EE}$ leukemia remarkably well, which translated into a significantly improved survival (Fig 7G and H).

## Discussion

Our results obtained with the R178E cooperativity mutation in different mouse cancer models illustrate that a p53 mutant can be as inefficient as a p53 knock-out allele in preventing tumor development and yet retain residual apoptotic activity. Taken together, this indicates that residual apoptotic p53 functions on their own are unable to counteract tumorigenesis, are therefore not efficiently counter-selected, and can be retained by mutant p53 during tumor evolution. Importantly, these residual cytotoxic activities can be triggered by chemotherapeutics to induce a survival benefit, especially when mutant p53 is constitutively or pharmacologically stabilized. Superior chemotherapy responses in p53-mutated versus p53-deficient tumors challenge the current view that a p53 missense mutation invariably signals a worse prognosis than a gene deletion.

p53WT is capable of inducing cell death through multiple pathways including apoptosis, necrosis, and ferroptosis. The cytotoxic activity of p53EE is apoptotic in nature as shown by caspase cleavage (Figs 3E and 4F and H), phosphatidylserine externalization (Figs 4A and 5C and H), and blockage by caspase inhibitors, but not ferroptosis inhibitors (Fig 5B). p53WT induces apoptosis by transcriptional upregulation of pro-apoptotic target genes such as PUMA, NOXA, and BAX in combination with non-transcriptional mechanisms mediated through protein–protein interactions with Bcl-2 family members, which lowers the threshold for engaging the mitochondrial apoptosis pathway (Mihara *et al*, 2003; Chipuk *et al*, 2004; Leu *et al*, 2004; Le Pen *et al*, 2016). The lack of detectable DNA binding and target gene regulation by p53EE suggested that its pro-apoptotic activity relies primarily on the non-transcriptional pathway, and we demonstrate that p53EE efficiently localizes to the mitochondria (Figs 4I and 5E and F), interacts with Bcl-2 family members (Fig 5F), and primes cells to mitochondrial depolarization by BH3 peptides (Fig 5G). In addition, we provide evidence that p53EE, similar to other p53 mutants, interferes with Nrf2 transcriptional activity, which is critical not only for ROS defense, but also for the respiratory function of mitochondria, their biogenesis, and integrity (Dinkova-Kostova & Abramov, 2015). It is therefore tempting to speculate that direct activities of p53 at the mitochondria and indirect effects through Nrf2-inhibition synergistically prime the mitochondria and lower the apoptotic threshold. Nevertheless, we cannot formally exclude that p53EE under conditions of maximal stimulation induces pro-apoptotic target genes to a degree that is below our detection limit. As embryonic development is very sensitive to deregulated p53 activity, residual transcriptional activity of p53EE could in principle explain the *Trp53$^{EE/EE}$;Mdm2$^{-/-}$* embryonic lethality. And indeed, it was shown that a single hypomorphic *Trp53$^{neo}$* allele (*Trp53$^{neo/-}$*) that shows ~7% activity is compatible with *Mdm2$^{-/-}$* embryonic development, whereas *Trp53$^{neo/neo}$* homozygosity, yielding ~16% activity, already triggers embryonic lethality (Wang *et al*, 2011). However, deregulated p53 can cause embryonic lethality by multiple pathways including apoptosis, ferroptosis, or cell growth arrest (Jiang *et al*, 2015; Moyer *et al*, 2017). Whether *Trp53$^{neo/neo}$;Mdm2$^{-/-}$* embryos die by apoptosis was not reported. In general, low p53 levels or reduced p53 DNA binding cooperativity primarily activates homeostatic survival functions, while only strongly elevated and prolonged expression levels shift the cellular response to apoptosis (Chen *et al*, 1996; Vousden & Lu, 2002; Vousden & Lane, 2007; Schlereth *et al*, 2013; Timofeev *et al*, 2013). This is mechanistically explained by higher affinity and torsionally more flexible p53 response elements in target genes regulating, for example, cell proliferation (Murray-Zmijewski *et al*, 2008; Riley *et al*, 2008; Schlereth *et al*, 2010b, 2013; Jordan *et al*, 2012) and a higher target gene expression threshold for triggering apoptosis (Kracikova *et al*, 2013). If p53EE regains residual transcriptional activity upon massive stimulation, it would be difficult to envision why this resulted in apoptosis instead of cell cycle arrest (Figs 3–5). Last but not least, the charge inversion associated with the arginine to glutamic acid substitution in p53EE might affect protein–protein interactions that are responsible for its pro-apoptotic activity. However, the same pro-apoptotic activity is seen for the charge-neutralizing R181L mutation (Fig 5H), excluding the negatively charged glutamic acid residue as the underlying cause. Altogether, we therefore favor the hypothesis that p53EE triggers apoptosis primarily through non-transcriptional mechanisms.

From a clinical perspective, it will be interesting to investigate which p53 mutants in cancer patients display residual apoptotic activity, as p53EE is not a naturally occurring mutation. Since

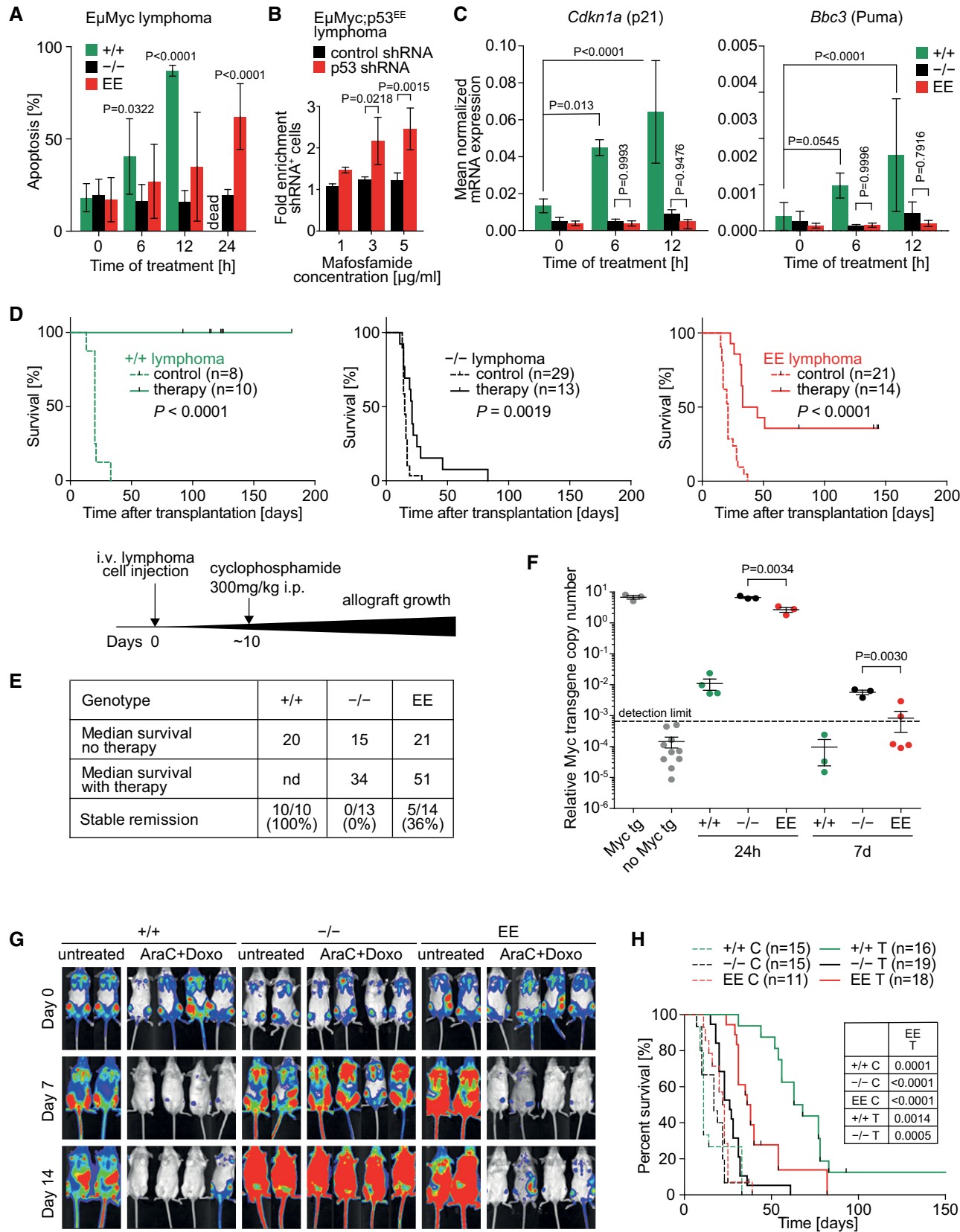

Figure 7.

**Figure 7. p53EE confers survival benefit under chemotherapy.**

A   Apoptosis in Eµ-Myc lymphoma cells of indicated genotype was measured by flow cytometry (FITC-VAD-FMK staining) at indicated time points after treatment with 3 µg/ml MAF. Significance was calculated versus untreated. $n \geq 5$.

B   Enrichment of p53 shRNA-expressing Eµ-Myc;p53$^{EE}$ lymphoma cells under MAF treatment. Non-silencing control shRNA served as control. $n = 3$.

C   mRNA expression of p53 target genes was measured relative to β-actin by RT–qPCR following treatment of Eµ-Myc lymphoma cells of indicated genotype with 3 µg/ml MAF. $n \geq 3$.

D, E   Kaplan–Meier survival plots for cyclophosphamide-treated versus untreated control mice transplanted with Eµ-Myc lymphoma cells of indicated genotype at day 0. Treatment started ~10 days later when peripheral lymph nodes became palpable. The time course of the experiment is illustrated in a scheme. (E) Summary of data from (D). Survival differences were analyzed using the log-rank (Mantel–Cox) test.

F   Residual disease in the spleen of mice from (D) was quantified 24 h and 7 days after therapy using a quantitative PCR assays with primers specific to the Eµ-Myc transgene (present only in lymphoma cells). Shown is the copy number of the Eµ-Myc transgene relative to a control locus present in all cells. Untreated Eµ-Myc mice (Myc tg) were used as positive control. Non-transgenic mice (no Myc tg) served as negative controls to define the detection limit. Significance was tested by *t*-test (two-sided, unpaired).

G   Representative bioluminescence imaging (BLI) pictures of mice transplanted with AE9+Nras AML cells. Day 0 indicates the start of combination therapy with cytarabine (AraC) + doxorubicin (Doxo).

H   Kaplan–Meier survival plots for animals from (G). Statistical significance of differences calculated by log-rank (Mantel–Cox) test between treated Eµ-Myc;p53$^{EE/EE}$ AML and all other groups is indicated in the table. C, untreated control; T, therapy (AraC+Doxo).

Data information: All data are shown as mean ± SD. Significance was tested by 2-way ANOVA with Sidak's multiple comparisons test unless indicated otherwise.

protein–protein interactions with Bcl-2 family members that account for non-transcriptional apoptosis by p53WT (Mihara *et al*, 2003; Chipuk *et al*, 2004; Leu *et al*, 2004; Le Pen *et al*, 2016) are disrupted by many cancer-derived p53 mutations, such mutations are believed to be "dual hits" which simultaneously inactivate both DNA binding-dependent and non-transcriptional mechanisms of p53-triggered apoptosis (Mihara *et al*, 2003; Tomita *et al*, 2006). However, this has only been shown for a small subset of mainly hot-spot mutants and cannot be directly extrapolated to the entire, functionally and structurally highly diverse, spectrum of > 2,000 distinct p53 mutations observed in cancer patients. The p53EE mutant provides evidence that transcriptional and non-transcriptional apoptotic functions can be genetically separated. Remarkably, the selective loss of p53 DNA binding in the presence of intact non-transcriptional apoptotic activity is just as efficient to promote tumorigenesis as the complete loss of p53 (Fig 6A, D and E), suggesting that the non-transcriptional activity of p53 is insufficient to prevent tumor development on its own. Of note, this does not exclude a supportive role of mitochondrial p53 functions for transcription-dependent apoptosis and is compatible with the model of mitochondrial priming by p53 that sensitizes to p53-induced target genes encoding the BH3-only proteins Puma and Noxa (Chipuk *et al*, 2005; Le Pen *et al*, 2016).

While the most common hot-spot mutants are thermodynamically destabilized and therefore structurally denatured, many non-hot-spot mutants retain the ability to regulate some *bona fide* p53 target genes like p21 (Ludwig *et al*, 1996; Rowan *et al*, 1996; Campomenosi *et al*, 2001). The ability to bind p53 response elements in target promoters implies sufficient structural stability to engage also in p53WT-like protein–protein interactions. We therefore anticipated that non-transcriptional apoptotic activities are retained primarily among non-hot-spot mutants, especially those with only a partial loss of DNA binding activity (Campomenosi *et al*, 2001). Several cooperativity mutants that together account for an estimated 34,000 cancer cases per year (Leroy *et al*, 2014) have selectively lost the ability to trans-activate pro-apoptotic target genes (Ludwig *et al*, 1996; Schlereth *et al*, 2010a). Here, we have examined the R181L mutant that has been identified as both a somatic and germline mutation in cancer patients. Consistent with previous reports that R181L has a selective apoptosis defect (Ludwig *et al*, 1996; Schlereth *et al*, 2010a), we failed to detect apoptosis upon R181L overexpression (Fig 5H). However, just like with p53EE, genotoxic doxorubicin treatment

revealed substantial pro-apoptotic activity, supporting the concept that a tumorigenic p53 mutant like R181L can retain residual apoptotic functions and support chemotherapy responses.

Intriguingly, the R175P mutant, which resembles the R181L mutant in its selective loss of pro-apoptotic target gene regulation (Ludwig *et al*, 1996; Rowan *et al*, 1996; Liu *et al*, 2004), failed to enhance apoptosis under identical conditions of doxorubicin treatment (Fig 5H). A possible explanation is that R175 is crucial for the structural integrity of the L2-L3 loop which interacts not only with the minor groove of DNA response elements but also with Bcl-2 family members (Cho *et al*, 1994; Bullock *et al*, 2000; Mihara *et al*, 2003; Tomita *et al*, 2006; Hagn *et al*, 2010). In contrast, R181 is solvent-exposed and R181 mutations do not affect the DBD 3D structure (Dehner *et al*, 2005). It is therefore tempting to speculate that in particular, cooperativity mutants of the H1 helix, which interfere with DNA binding but do not affect the interaction interface with DNA and Bcl-2 family members, have retained non-transcriptional apoptotic activity. A more detailed systematic analysis of the more than 2,000 different cancer mutants is needed to validate this hypothesis and possibly reveal further cancer mutants with a similar phenotype.

Recent studies have revealed that tumors can be addicted to pro-survival GOF activities of mutant p53 and respond to mutant p53 depletion with tumor regression (Alexandrova *et al*, 2015). Mutant p53-destabilizing drugs, such as the Hsp90 inhibitor ganetespib, are therefore considered a promising approach for treatment of p53-mutated tumors, and a first clinical trial combining chemotherapy with ganetespib for p53-mutated ovarian cancer patients has been initiated (Alexandrova *et al*, 2015; Bykov *et al*, 2018; Sabapathy & Lane, 2018). However, mouse models for the R246S (equivalent to human R249S) and humanized G245S mutants have shown that p53 mutations can be highly tumorigenic without exhibiting appreciable GOF activity (Lee *et al*, 2012; Hanel *et al*, 2013). The p53EE mutant does not significantly decrease the latency of tumor development or enhance the incidence of metastatic tumors compared to p53 knock-out mice (Fig 6A; Appendix Tables S1 and S2). There was a trend for increased incidence of non-lymphoid tumors in homozygous p53EE mice, but this was not statistically significant (Fig 6B), leading us to conclude that p53EE does not exhibit the strong GOF activities observed for some hot-spot mutants. When tumors are not dependent on a mutant p53 GOF, its destabilization is expected to be therapeutically ineffective. More importantly, however, our data

imply that a combination of chemotherapy with mutant p53 destabilizing drugs could even turn out counterproductive or dangerous when chemotherapy-sensitizing effects are abolished by removal of the p53 mutant. In support of this, degradation of p53EE by ganetespib did not result in tumor cell death by itself and effectively counteracted the pro-apoptotic effect of combined doxorubicin and Nutlin treatment (Fig 4H).

Instead, our data suggest that tumors which contain a p53 mutant with residual apoptotic activity might require the opposite strategy. Chemotherapy responses of p53EE cells were enhanced in combination with Nutlin-3a or other Mdm2 inhibitors (Fig 4D and E). Enforced overexpression of the human p53EE (R181E) or the cancer mutant R181L strongly sensitized human lung cancer cells to chemotherapy-induced apoptosis, and this was further boosted by Nutlin-3a (Fig 5H). Similarly, lymphoma and leukemia cells with massive constitutive stabilization of p53EE displayed superior chemotherapy responses to p53-deficient tumor cells resulting in improved animal survival (Fig 7). Mutant p53-stabilizing therapies with Mdm2 inhibitors might therefore support chemotherapeutics to trigger the residual cytotoxic activities that have not been efficiently counter-selected during tumorigenesis. In fact, the first clinical studies with Mdm2 inhibitors have observed clinical responses also in a few patients with p53 mutations (Andreeff *et al*, 2016). Even though Mdm2 inhibitors were originally designed for patients with wild-type p53 tumors, they might eventually offer therapeutic benefit also for some p53-mutated cancer patients. Of course, a caveat to the treatment with mutant p53 stabilizing compounds is that pro-metastatic or pro-survival GOF activities might be boosted. The most promising therapy strategy for p53-mutated tumor patients will therefore strongly depend on the functional properties of the p53 mutant. Although it remains to be seen, which and how many p53 mutants have retained apoptotic activity during tumor evolution, our data call for more comprehensive investigations into the functional diversity of p53 mutations to make p53 mutation status more useful for clinical decision making.

# Materials and Methods

### Animals

For generation of the $Trp53^{LSL-R178E}$ knock-in mouse, we used the targeting vector for the $Trp53^{LSL-R172H}$ mouse, which was kindly provided by Tyler Jacks (Olive *et al*, 2004). The vector was modified by QuikChange Multi Site-Directed Mutagenesis (Stratagene) to carry only the mutation GCG->CTC in exon 5 of $Trp53$, resulting in a Arg->Glu substitution at codon 178 (Fig EV1A). The complete targeting vector was verified by Sanger sequencing. The linearized targeting vector was transfected by electroporation of 129/SvEv (129) embryonic stem cells. After selection with G418 antibiotic, surviving clones were expanded for PCR analysis to identify recombinant ES clones and verify the presence of the R178E mutation by Sanger sequencing. ES cell clones were further validated by Southern blot for correct 5′ and 3′ homologous recombination using *Ssp*I digest in combination with the 3′ probe and *Xba*I digest with the 5′ probe (Fig EV1B). A correctly targeted ES cell clone was microinjected into C57BL/6 blastocysts. Resulting chimeras with a high percentage agouti coat color were mated to wild-type 129/SvEv

mice to generate F1 heterozygous offspring. Tail DNA was analyzed as described below from pups with agouti coat color (Fig EV1C and D). Additional mutations were excluded by sequencing of all exons and exon–intron boundaries in the $Trp53$ gene. ES cell cloning, blastocyst injection, and breeding of chimeras for germline transmission were done at inGenious Targeting Laboratory (Stony Brook, USA). Knock-in mice were kept on a pure 129Sv background.

In 129S/Sv-$Trp53^{LSL-R178E}$ mice, expression of the $Trp53$ gene is blocked by a transcriptional stop cassette, flanked by loxP sites (lox-stop-lox, LSL). The homozygous $Trp53^{LSL-R178E/LSL-R178E}$ mice lack p53 protein expression. $Trp53^{LSL-R178E/LSL-R178E}$ mice, and embryonic fibroblasts isolated from these mice, were used as isogenic p53-null ($Trp53^{-/-}$) controls in all experiments. For generation of knock-in mice expressing the p53EE mutant protein, we crossed heterozygous $Trp53^{+/LSL-R178E}$ mice with Prm-Cre transgenic animals (129S/Sv-Tg (Prm-cre)58Og/J; Jackson Laboratory). In double transgenic males, the Prm1-Cre allele mediates excision of the LSL cassette in the male germline. These males were used for breeding with wild-type 129/Sv females to obtain constitutively recombined $Trp53^{+/R178E}$ mice, which were then intercrossed to generate homozygous knock-in and wild-type $Trp53^{+/+}$ (control) animals. In aging cohorts, mice were monitored on the daily basis and sacrificed when they reached objective criteria for a humane endpoint that were defined before the onset of the experiment.

To study lethality of $Mdm2$ knock-out, we used heterozygous $Mdm2^{\Delta7-9}$ (Mdm2<tm1.2Mep>) mice that lack exons 7–9. These mice were obtained from the NCI Mouse Repository (Frederick, USA) and crossed with $Trp53^{EE/EE}$ homozygotes to obtain double heterozygous $Trp53^{+/EE};Mdm2^{+/\Delta7-9}$ animals for intercrossing. Genotypes of mice, isolated embryos, and embryonic tissues were identified by PCR. Primer sequences are listed in Table EV1.

All mouse experiments were performed in accordance with the German Animal Welfare Act (Deutsches Tierschutzgesetz) and were approved by the local authorities.

### Murine lymphoma model

For generation of Burkitt-like B-cell lymphoma, heterozygous females with different p53 genotypes were crossed to transgenic B6.Cg-Tg(IghMyc)22Bri/J males (Jackson Laboratory; Adams *et al*, 1985). Primary lymphomas were obtained from Eμ-Myc mice with p53$^{+/+}$, p53$^{+/-}$, and p53$^{+/EE}$ genotypes, and 500,000 cells were transplanted into syngeneic recipients via tail vein as described in the literature (Schmitt *et al*, 2002a). After lymphomas became palpable as enlarged peripheral lymph nodes, mice were separated into two groups—one group was treated with a single intraperitoneal dose of 300 mg/kg cyclophosphamide (Cell Pharm, Hannover, Germany), and another group was left untreated. Samples were collected 6 h, 24 h, and 7 days after therapy. In both groups, mice were monitored on the daily basis for up to a maximum of 4 months and sacrificed when they reached objective criteria for a humane endpoint that were defined before the onset of the experiment.

### Murine acute myeloid leukemia (AML) model

Primary AML with different p53 genotypes was generated as described (Zuber *et al*, 2009). In brief, fetal liver cells were isolated from p53$^{+/+}$, p53$^{-/-}$, and p53$^{EE/EE}$ embryos at E14-16. Retroviral

plasmids encoding the AML1/ETO9a fusion oncogene (co-expressed with GFP) and NRas$^{G12D}$ oncogene (co-expressed with firefly luciferase) were kindly provided by J. Zuber (Zuber *et al*, 2009). Recombinant retroviruses were packaged in Platinum-E cells (Cell Biolabs). After four rounds of infection with the two retroviral vectors mixed 1:1, transfection efficiency was analyzed by flow cytometry. 24–48 h after last infection, 1 million cells were transplanted intravenously into lethally irradiated (7 Gy) 129X1/SvJ albino recipients. F1 hybrids from breeding 129X1/SvJ/albino X C57B6/albino mice were used as secondary recipients for chemotherapy. After primary recipients developed advanced leukemia with accumulation of malignant myeloid progenitors in bone marrow and infiltration of extramedullary tissues such as spleen, AML cells were isolated, immunophenotyped, and transplanted into sublethally irradiated (3.5 Gy) secondary recipients. Disease development was monitored by bioluminescence imaging (BLI). BLI was performed using an IVIS 100 Imaging System (Xenogen) under isoflurane anesthesia, 5 min after intraperitoneal injection of 200 μl D-luciferin (15 mg/ml in PBS, BioVision). After first detection of clear signal in bones and initial spleen infiltration, mice were separated into control and therapy groups and the latter was subjected to a standard chemotherapy protocol containing cytarabine (Cell Pharm, Hannover, Germany) and doxorubicin (Cell Pharm, Hannover, Germany): 3 days of cytarabine 100 mg/kg + doxorubicin 3 mg/kg intraperitoneally, followed by 2 days of cytarabine only. Mice under therapy were provided with drinking water containing 120 mg/l ciprofloxacin (Bayer) and 20 g/l glucose (Sigma). Therapy response was monitored by BLI. Mice were monitored on the daily basis and sacrificed when they reached objective criteria for a humane endpoint that were defined before the onset of the experiment.

## Cell culture and gene transfer

Human lung cancer cell lines (H1299, H460) were obtained from the American Tissue Collection Center (ATCC) and authenticated by short tandem repeat analysis at the Leibniz Institute DSMZ—German Collection of Microorganisms and Cell Cultures, Braunschweig, Germany. Cells were maintained in high-glucose Dulbecco's modified Eagle's medium (DMEM) supplemented with 10% fetal bovine serum (FBS, Sigma-Aldrich), 100 U/ml penicillin, 100 μg/ml streptomycin (Life Technologies), and 1 μg/ml amphotericin B (Sigma) at 37°C with 5% $CO_2$. Primary MEFs were isolated from E12.5-13.5 mouse embryos and amplified under low oxygen conditions (3% $O_2$); their gender is not available. Eμ-Myc lymphoma cell lines were maintained in B-cell medium: 1:1 mixture of DMEM and Iscove's modified Dulbecco's medium (IMDM) supplemented with 20% FBS, 100 U/ml penicillin/streptomycin, 50 μM 2-mercaptoethanol, and 1 ng/ml mIL-7 (ImmunoTools). Cells were cultured on a feeder layer of 30 Gy-irradiated NIH 3T3 cells. Nutlin-3a (Sigma) was used at 10 μM, RG7112 (MedChemExpress) at 5 μM, RG7388 (MedChemExpress) at 8 μM, MI773 (Selleckchem) at 10 μM, and Mafosfamide (Santa Cruz) at 1–5 μg/ml as indicated. Hydrogen peroxide (Sigma) was used at 50–800 μM.

## Plasmids and gene transfer

Transfections and viral infections were performed as described (Timofeev *et al*, 2013). For production of lentiviruses, helper plasmids pMD2.g (Addgene plasmid #12259) and psPAX2 (Addgene plasmid #12260) were used. For Tet-inducible gene knockdown with shRNAs, we used the retroviral vector TtRMPVIR (Addgene plasmid #27995), and for Tet-inducible gene expression, we used the lentiviral vector pInducer20 (Addgene plasmid #44012). For expression of mutant p53, the cDNA for murine or human p53 was cloned into pInducer20 using the Gateway® System (Invitrogen). MEFs were immortalized by retroviral transduction with pMSCVhygro-E1A.12S or pMSCVneo-E1A.12S (Timofeev *et al*, 2013). For studies on Ras-induced senescence, MEFs were transduced with MSCVhygro-HRas$^{G12V}$ (Timofeev *et al*, 2013).

## CRISPR-Cas9 gene editing

For the generation of p53 knock-out cells using CRISPR-Cas9 gene editing, oligos encoding sgRNAs targeting the mouse *Trp53* gene or GFP as a control were annealed and cloned by Golden Gate cloning into BsmBI-digested lentiCRISPR vector (Addgene plasmid #49535). The presence of indels in the targeted *Trp53* gene locus was confirmed by T7 endonuclease assay. In brief, infected cells were collected, genomic DNA was isolated, and the fragment of interest was PCR-amplified using primers flanking the predicted CRISPR-Cas9 cleavage site. The PCR product was purified using the QIAquick PCR Purification Kit (Qiagen), DNA was denatured, reannealed in 1× NEB2 buffer (NEB), digested with 5 U T7 endonuclease I (NEB) for 20 min at 37°C, and analyzed on a 2% agarose gel. Knock-out of p53 in the pool of cells was confirmed by p53 immunofluorescence staining, Sanger sequencing of the targeted *Trp53* region, and InDel analysis using the TIDE algorithm (Brinkman *et al*, 2014). sgRNA and primer sequences are listed in Table EV1.

## Cell viability and proliferation assays

Viability of cells after irradiation or drug treatment was assessed using CellTiter-Glo Luminescent Cell Viability Assay (Promega) as described (Timofeev *et al*, 2013). For quantitative real-time monitoring of cell proliferation, we performed automated time-lapse microscopy using the IncuCyte S3 (Essen BioScience). Cells were seeded in triplicates on 96-well plates to reach a starting confluence of 10–30% after 24 h. Following treatment with Nutlin ± doxorubicin, the plates were imaged for 48 h with a time interval of 2 h. Confluence was calculated using the instrument's IncuCyte Zoom 2017A software. For assessment of long-term proliferative capacity of primary MEFs (Fig EV4A), we used a modified 3T3 protocol: 200,000 cells from passage 1–2 were plated to each well of a 6-well plate. Every 3 days, cells were harvested and counted and 200,000 cells were replated. When the total number of cells was lower than 200,000, all cells were replated. Population doubling was calculated as the $\log_2$ of the ratio $N_{px+1}/N_{px}$, where $N_{px}$ is the total number of cells at passage X. For proliferation assay with CRISPR-Cas9-edited MEFs (Fig EV3E) and for proliferation at normal (21%) and low (3%) oxygen conditions (Fig EV3H), 500,000 cells were replated every 3 days on 10-cm cell culture dishes. Under these conditions, proliferation of p53$^{EE/EE}$ MEFs declined more rapidly than in Fig EV4A and eventually resulted in deterioration of cell cultures.

## Cellular fractionation and western blotting

For immunoblotting, cells were lysed in RIPA buffer (0.1% SDS, 50 mM Tris–HCl at pH 7.4, 150 mM NaCl, 1 mM EDTA, 1% Na-deoxycholate) supplemented with protease inhibitor cocktail (Roche). To prepare nuclear and cytosolic fractions, cells were collected and resuspended in 2–3 volumes of Buffer A (10 mM HEPES pH 7.9, 10 mM KCl, 0.1 mM EDTA, 0.1 mM EGTA, 1 mM β-mercaptoethanol) supplemented with protease inhibitors, and incubated for 10 min on ice before adding 10% NP-40 to a final concentration of 0.25%. Cells were passed through a 27-G needle 5–10 times using a 1-ml syringe. Nuclei were pelleted by centrifugation (500 ×g, 10 min). The cytoplasmic fraction was collected and re-centrifuged for 10–15 min. Nuclear fractions were washed three times in 5–10 volumes of Buffer A and pelleted by centrifugation. Nuclei were lysed with 1–2 volumes of Buffer C (20 mM HEPES pH 7.9, 400 mM NaCl, 1 mM EDTA, 1 mM EGTA, 1 mM β-mercaptoethanol) and soluble fraction collected by centrifugation (10,000 ×g, 10 min). Mitochondrial fractions were isolated using Mitochondria Isolation Kit for Cultured Cells (Thermo Scientific) as described by the manufacturer. For Western blotting, 20–50 μg of total protein was resolved on 4–12% NuPAGE polyacrylamide gels (Invitrogen). After wet transfer to Hybond P nitrocellulose membrane (GE Healthcare), antigens were detected using the following antibodies: anti-cleaved caspase-3 (#9661, Cell Signaling, 1:500), anti-p53 (NCL-p53-505, Leica Microsystems, 1:2,000), anti-MDM2 (SMP14, #sc-965, Santa Cruz Biotechnology [SC], 1:200), anti-PCNA (PC10, #sc-56, 1:1,000), anti-Bak (At8B4, Abcam, 1:250), anti-Tom20 (FL-145, #sc-11415, 1:100), anti-cleaved PARP (#9541, Cell Signaling, 1:500), anti-cleaved Lamin A small subunit (#3035, Cell Signaling, 1:500), anti-H-Ras (C-20, #sc-520, Santa Cruz Biotechnology, 1:100), anti-E1A (M-73, #sc-25, Santa Cruz Biotechnology, 1:500), and anti-β-actin (AC-15, #ab6276, Abcam, 1:10,000). Detection was performed with secondary anti-mouse or anti-rabbit IgG-HRP (GE Healthcare, 1:5,000) and SuperSignal ECL Kit (Thermo Fisher).

## Immunohistochemistry and immunofluorescence

For histology and immunohistochemistry (IHC), formalin-fixed samples were embedded in paraffin and 5-μm sections were mounted to glass slides and processed as described (Timofeev et al, 2013). Apoptosis was detected using the DeadEnd™ Colorimetric TUNEL System (Promega) or antibodies against cleaved caspase-3 (#9661, Cell Signaling, 1:100). Other antibodies used for IHC were anti-p53 (NCL-p53-505, Leica Microsystems, 1:1,000) and anti-BrdU (BU1/75(ICR1), #OBT0030G, 1:100).

## Proximity ligation assays

In situ interactions were detected by the proximity ligation assay kit Duolink (DPLA probe anti-rabbit minus, DPLA probe anti-mouse plus (Sigma-Aldrich, St. Louis, MO, USA); Detection Kit Red, Sigma-Aldrich). The DPLA probe anti-rabbit minus binds to the p53 antibody, whereas the DPLA probe anti-mouse plus binds to the antibody against the probable interaction partner, respectively. The Duolink proximity ligation assay secondary antibodies generate only a signal when the two DPLA probes have been bound, which only takes place if both proteins are closer than 40 nm, indicating their interaction. Paraformaldehyde-fixed FFPE sections were pre-incubated with blocking agent for 1 h. After washing in PBS for 10 min, primary antibodies were applied to the samples. Incubation was done for 1 h at 37°C in a pre-heated humidity chamber. Slides were washed three times in PBS for 10 min. DPLA probes detecting rabbit or mouse antibodies were diluted in the blocking agent in a concentration of 1:5 and applied to the slides followed by incubation for 1 h in a pre-heated humidity chamber at 37°C. Unbound DPLA probes were removed by washing two times in PBS for 5 min. The samples were incubated with the ligation solution consisting of Duolink Ligation stock (1:5) and Duolink Ligase (1:40) diluted in high-purity water for 30 min at 37°C. After ligation, the Duolink Amplification and Detection stock, diluted 1:5 by the addition of polymerase (1:80), was applied to the slides for 100 min. Afterward, the slides were incubated with DAPI for the identification of nuclei. After the final washing steps, the slides were dried and coverslips were applied. Quantification was done with the Duolink Image Tool v1.0.1.2 (Olink Bioscience, Uppsala, Sweden). The signal threshold was adjusted to 100 and the pixel size for spot detection to 5 pixels for each picture.

## Flow cytometry

For flow cytometry analysis, an Accuri C6 Flow Cytometer (BD Biosciences) was used. Labeling of S-phase cells with BrdU and processing for FACS were done as described (Timofeev et al, 2013) using anti-BrdU Alexa Fluor 488 antibodies (BD Biosciences #347580). For analysis of apoptosis, annexin V-APC (MabTag) and CaspGLOW Staining Kit (BioVision) were used according to manufacturer's protocols. Cell cycle profiles and viability of cells were assessed using staining with propidium iodide (PI) in permeable and non-permeable conditions as described earlier (Timofeev et al, 2013). To analyze mitochondrial ROS levels, cells were stained for 15 min on plates with 3 μM MitoSOX free radical sensor (Thermo Fisher) in HBSS buffer (Sigma), collected with trypsin, washed with PBS, and analyzed by flow cytometry.

Eμ-Myc;p53$^{EE}$ lymphoma cells were transduced with TtRMPVIR retroviral vectors (Zuber et al, 2011) expressing a p53-targeting (shp53.814, Dickins et al, 2005) or non-silencing control shRNA coupled to RFP. Knockdown of p53 was induced with 1 μg/ml doxycycline. After 48 h of doxycycline pre-treatment, cells were treated with 1–5 μg/ml mafosfamide (Santa Cruz) for 24 h, washed, and left to recover for 48 h. Enrichment of RFP-expressing cells, i.e., cells efficiently expressing shRNA, in treated versus untreated cultures was analyzed with flow cytometry.

## Metabolism analysis

Mitochondrial metabolism was assessed using the Seahorse XF Cell Mito Stress Test Kit (Agilent) on the Seahorse XFe96 instrument according to manufacturer's protocols. Cells were treated for 24 h with 0.5 μg/ml doxycycline to induce p53 expression. A total of 20,000 cells were plated in 96-well XF cell culture microplates and let to adhere for 5 h before measurements were performed. Data were analyzed using MS Excel templates provided by the manufacturer.

## Electrophoretic mobility shift assays

Electrophoretic mobility shift assays (EMSAs) were performed in 20 μl reaction volume containing 20 mM HEPES (pH 7.8), 0.5 mM

EDTA (pH 8.0), 6 mM MgCl₂, 60 mM KCl, 0.008% Nonidet P-40, 100 ng anti-p53 antibody (Pab421), 1 mM DTT, 120 ng salmon sperm DNA, 1 µl glycerol, 20,000 cpm of [³²P]-labeled double-stranded oligonucleotide, and either 5 µl of *in vitro* translated protein or 5 µg of nuclear extracts from Nutlin-treated MEFs. After 30 min incubation at room temperature, reaction mixtures were subjected to electrophoresis on a 3.5% native polyacrylamide gel (37.5:1 acrylamide/bisacrylamide) in a Tris-borate-EDTA buffer at 125 V for 90 min at room temperature. 10 pmoles of competitor were added to control the specificity of DNA binding. For supershift analysis, 1 µg of anti-p53 antibody (FL393, Santa Cruz) was added. DNA–protein complexes were revealed with X-ray films (Kodak).

## Chromatin immunoprecipitation

Primary MEFs were treated for 16 h with 10 µM Nutlin or 0.1% DMSO as control. Cells were fixed on plates with 0.88% paraformaldehyde (PFA) for 10 min at room temperature and quenched by adding glycine to a final concentration of 54 mM for 5 min. Cells were washed twice with ice-cold PBS and scraped off the plate with PBS supplemented with proteinase inhibitors (Roche). Cell pellets were lysed in SDS lysis buffer (1% SDS, 10 mM EDTA, 50 mM Tris–HCl pH 8.1) supplemented with protease inhibitors (2 × 10⁷ cells/ml lysis buffer) and sonicated using the Bioruptor® Sonication System (Diagenode) to obtain 300–1,000 bp DNA fragments. The shearing efficiency was controlled using agarose gel electrophoresis. After centrifugation (10,000 *g*, 10 min, 20°C), 100 µl sheared chromatin was diluted 1:10 with dilution buffer (0.01% SDS, 1.1% Triton X-100, 1.2 mM EDTA, 16.7 mM Tris–HCl pH 8.1, 167 mM NaCl) and pre-cleared for 1 h with 50 µl Protein G Sepharose (50% slurry in 20% ethanol) beads (GE Healthcare) at 4°C. The p53 protein was immunoprecipitated overnight at 4°C with 2.5 µg anti-p53 antibody (FL393, Santa Cruz), and normal rabbit IgG (Santa Cruz) was used as the control. The protein complexes were pulled down for 4 h at 4°C with 50 µl Protein G Sepharose beads. Beads were washed with Low Salt Immune Complex Wash Buffer (0.1% SDS, 1% Triton X-100, 2 mM EDTA, 20 mM Tris–HCl pH 8.1, 150 mM NaCl), then with High Salt Immune Complex Wash Buffer (0.1% SDS, 1% Triton X-100, 2 mM EDTA, 20 mM Tris–HCl pH 8.1, 500 mM NaCl), with LiCl Immune Complex Wash Buffer (0.25 M LiCl, 1% IGEPAL-CA630, 1% deoxycholic acid (sodium salt), 2 mM EDTA, 10 mM Tris–HCl pH 8.1), and finally twice with TE (10 mM Tris–HCl pH 8.1, 1 mM EDTA). Protein/DNA crosslinks were eluted twice for 15 min at 20°C in 100 µl elution buffer (0.1 M NaHCO3, 1% SDS). 1% input stored at −20°C was treated in the same way. Crosslinking was reverted upon overnight incubation at 65°C in elution buffer supplemented with 200 mM NaCl followed by RNase A digestion at 37°C for 30 min, addition of 40 mM Tris–HCl pH 6.5 and 10 mM EDTA, proteinase K digestion for 2 h at 55°C, and inactivation of proteinase K at 99°C for 10 min. DNA was purified using the QIAquick PCR Purification Kit (Qiagen), and DNA concentration was measured with the Qubit dsDNA HS reagent (Molecular Probes).

## ChIP-seq and RNA-seq

ChIP-seq libraries were prepared from purified ChIP DNA with the MicroPlex Library Preparation Kit (Diagenode) according to the manufacturer's instructions. For RNA-seq, RNA quality was assessed using the Experion RNA StdSens Analysis Kit (Bio-Rad). RNA-seq libraries were prepared from total RNA using the TruSeq Stranded mRNA LT Kit (Illumina) according to the manufacturer's instructions. Quality of sequencing libraries (for ChIP-seq and RNA-seq) was controlled on a Bioanalyzer 2100 using the Agilent High Sensitivity DNA Kit (Agilent). Pooled sequencing libraries were quantified with digital PCR (QuantStudio 3D, Thermo Fisher) and sequenced on the HiSeq 1500 platform (Illumina) in rapid-run mode with 50 base single reads.

## ChIP-seq data analysis

Reads were aligned to the *Mus musculus* genome retrieved from Ensembl revision 79 (mm10) with Bowtie 2.0.0-beta7 using the default parameter settings (Langmead & Salzberg, 2012). Lanes were deduplicated to a single duplicate read, keeping only these effective reads for further analysis. Peak calling was performed individually for each sample using a mixed input as background using MACS 1.4.0rc2 with default parameters for all samples (Zhang *et al*, 2008). To reduce the amount of false positives, peaks were filtered to those peaks showing a strong enrichment over background, i.e., peaks with a minimum of 50 effective foreground reads, not more than 50 effective reads in background, and showing at least a threefold increase in the normalized read counts compared to background. To enable comparison between the samples, tag counts were calculated and normalized to one million mapped reads (tags per million, TPM). The foreground–background ratio used to filter reported peaks was calculated on basis of TPMs in foreground versus TPMs in background. To compare samples, filtered ChIP-seq peaks from the Nutlin-treated p53⁺/⁺ sample were taken as the basis. Of these 472 peaks, we removed 4 peaks that overlap with non-specific peaks in Nutlin-treated p53⁻/⁻ MEFs, yielding 468 p53WT peaks. 2,000 bp spanning regions around all of these 468 peak regions were centered to the summit of the p53⁺/⁺ Nutlin signal (TPM). To allow comparability of the lanes, TPMs were plotted as heatmap for all lanes, and the maximum value was fixed to the 95th percentile of the p53⁺/⁺ Nutlin signal. To obtain a list of target genes, we annotated the gene with the closest corresponding transcription start site to each peak. Hypergeometric enrichment was performed using Fisher's exact test, based on MSigDB gene sets as reference and our target genes as query.

Motif identification: For *de novo* binding motif search, peaks from Nutlin-treated p53⁺/⁺ MEFs (filtered as stated above) were trimmed to 150 bp (peak middle +/−75 bp). These 468 regions were used for analysis with MEME-ChIP (version 4.12.0) using the default parameters (Machanick & Bailey, 2011). To identify motifs of known binding sites, CentriMo (version 4.12.0) analysis was performed on the same 150 bp regions, using the JASPAR CORE 5 (vertebrates) and UniPROBE (retrieved August 2011) databases as reference (Berger *et al*, 2006; Bailey & Machanick, 2012; Mathelier *et al*, 2014).

## RNA-seq data analysis

Reads were aligned to the *Mus musculus* genome retrieved from Ensembl revision 79 (mm10) with STAR 2.4. Tag counts were calculated and normalized to one million mapped exonic reads and gene length (FPKM). To generate the set of expressed genes, only genes with a minimum read count of 50 and a minimum FPKM of 0.3 were

kept. DESeq2 (version 1.8.2) was used to determine differentially expressed genes (Berger *et al*, 2006). Genes with a greater than twofold change according to DESeq2 analysis and a maximum FDR of 0.05 were considered as regulated. Gene set enrichment analysis (GSEA) was performed using Molecular Signatures Database (MSigDB) gene sets and GSEA2 software (version 2.1.0) from the Broad Institute (West *et al*, 2005; Liberzon *et al*, 2011, 2015). FPKM values of genes from the indicated gene sets were z-transformed and plotted as heatmaps.

### PCR and real-time PCR

For PCR genotyping, tail tips or tissues were lysed overnight at 55°C in PBND buffer (10 mM Tris–HCl pH 8.3, 50 mM KCl, 2.5 mM MgCl2, 0.45% NP-40, 0.45% Tween-20) supplemented with 8 U/ml proteinase K (AppliChem) and afterward inactivated at 95°C for 10 min. For LOH and residual disease detection in Eμ-Myc lymphoma samples, crude lysates were prepared as above and 50 ng of genomic DNA purified with peqGOLD Tissue DNA Mini Kit (PeqLab) was used as template for qPCR with primers specific for the EμMyc transgene and a control locus for normalization.

For reverse transcription–quantitative PCR (RT–qPCR), RNA was isolated from cells or tissue samples using the RNeasy Mini Kit (Qiagen) and cDNA was generated with the SuperScript VILO cDNA Synthesis Kit (Invitrogen). Gene expression was analyzed on a LightCycler 480 (Roche) using the ABsolute QPCR SYBR Green Mix (Thermo Scientific). Data were evaluated by the ΔΔCt method with β-actin as a housekeeping gene for normalization.

For mtDNA content analysis, genomic DNA was analyzed by quantitative PCR for the mitochondrial genes *mt-Nd4*, *mt-Co1*, *mt-Cyb*, and *mt-Nd2*, and Ct values were normalized to the nuclear gene *Trp53*.

Primer sequences are provided in Table EV1.

### Statistical analyses

Statistical differences between experimental groups analyzed at different time points or under different treatment conditions were calculated using one- or two-way ANOVA test, while correcting for multiple comparisons using Sidak's multiple comparisons test. A significance level of $P < 0.05$ was used throughout the study. Kaplan–Meier curves representing survival analyses were compared by the log-rank (Mantel–Cox) test. Multiple survival Kaplan–Meier curves were compared by ordinary ANOVA with Tukey's multiple comparisons test. Statistical analyses were performed using the Prism software package (GraphPad).

## Data availability

The datasets produced in this study are available in the following databases:

- RNA-seq data: EBI ArrayExpress E-MTAB-6774
- ChIP-seq data: EBI ArrayExpress E-MTAB-6793

**Expanded View** for this article is available online.

## Acknowledgements

We thank Ute Moll, Yinon Ben-Neriah, and members of the laboratory for helpful discussion and advice; and Tyler Jacks and Johannes Zuber for providing plasmids. We acknowledge Sigrid Bischofsberger, Angelika Filmer, Alexandra Schneider, Antje Grzeschiczek, Angela Mühling, Johanna Grass, Björn Geissert, and trainee Mikhail Moskalenko for excellent technical assistance; the Cell Metabolism Core Facility (Wolfgang Meissner) for performing Seahorse assays; and the Irradiation Core Facility (Rita Engenhart-Cabillic) for providing access to the X-RAD 320iX platform. This work was supported by grants from the Deutsche Forschungsgemeinschaft (DFG) (DFG STI 182/7-1; TI 1028/2-1), Deutsche Krebshilfe (# 111250, 111444, 70112623), Deutsche José Carreras Leukämie Stiftung e.V. (DJCLS R 13/08, 09 R/2018), Bundesministerium für Bildung und Forschung (BMBF 031L0063), and German Center for Lung Research (DZL).

## Author contributions

Conducting experiments: OT, BK, JS, MW, JN, SM, AN, EP, UW, AB, and KK; conceptual input and supervision: OT, BK, SE, SG, and TS; data analysis: OT, BK, MM, BL, UW, and TS; and project design and writing: OT and TS.

## Conflict of interest

The authors declare that they have no conflict of interest.

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
