## [Review Process File · The EMBO Journal]

Residual apoptotic activity of a tumorigenic p53 mutant improves cancer therapy responses

Oleg Timofeev, Boris Klimovich, Jean Schneikert, Michael Wanzel, Evangelos Pavlakis, Julia Noll, Samet Mutlu, Sabrina Elmshäuser, Andrea Nist, Marco Mernberger, Boris Lamp, Ulrich Wenig, Alexander Brobeil, Stefan Gattenlöhner, Kernt Köhler and Thorsten Stiewe

Review timeline:

Submission date:	25th Mar 2019
Editorial Decision:	2nd May 2019
Revision received:	13th Jun 2019
Editorial Decision:	25th Jul 2019
Revision received:	30th Jul 2019
Accepted:	5th Aug 2019

Editor: Daniel Klimmeck

Transaction Report:

1st Editorial Decision

2nd May 2019

Thank you for your interest and the submission of your manuscript (EMBOJ-2019-102096) to The EMBO Journal. Your manuscript has been sent to three referees for consideration, and we have received reports from all of them, which I enclose below.

As you will see, the referees acknowledge the potential high interest and novelty of your work, although they also express a number of issues that will have to be addressed before they can support publication of your manuscript in The EMBO Journal. In more detail, referee #2 is concerned that the mechanistic details by which ROS-induced senescence is triggered and related implications of metabolism are not sufficiently characterized in his/her view (Ref#2, pts.3,5; see also third comment referee #1). This referee also asks you to strengthen the relevance of your findings for patient stratification by investigating additional data (ref#2, pt.1). Referee #3 agrees in that clinical angle should be interrogated further to address potential therapeutic benefit.

I judge the comments of the referees to be generally reasonable and given their overall interest, we are in principle happy to invite you to revise your manuscript experimentally to address the referees' comments.

REFeree REPORTS:

Referee #1:

Previous studies from the authors' laboratory had identified specific residues on the p53 tumor suppressor that influence its ability to engage in cooperative sequence-specific DNA binding. In this manuscript, they utilize a new mouse model with a missense mutation within the endogenous p53 gene to provide three novel and interesting findings about p53 and its role in cancer. First, they clearly show that the ability of p53 to rescue the embryonic lethality of Mdm2 knockout mice is

distinct from its role in tumor suppression. This supports the notion that the role of p53 in development is different from that in cancer. Second, they provide compelling evidence to support the idea that mutant p53-driven tumors will be responsive to chemotherapy in tissues and under conditions in which p53-null tumors are not. This has translational potential and high significance for designing personalized therapies using p53 status as a guide. Third, the data herein firmly establish a role for transcriptional-independent effects of p53 in tumor suppression. This idea has been examined extensively in the past but these authors now provide genetic evidence to separate cytoplasmic from nuclear roles for p53.

The manuscript is clearly written. Multiple and diverse approaches are used to support the findings. Given the significance and innovation, the study is quite suitable for The EMBO Journal. There are three concerns that should be addressed before publication.

First, a notable finding is the observation of embryonic lethality with Mdm2 knockout by the p53EE mutant at E9.5. However, this is distinct from what is observed with p53 null embryos where lethality is observed at the blastocyst stage. The authors do not acknowledge or discuss this important difference. Data comparing the p53EE mutant embryos to p53 null embryos is essential and needs to be included and discussed.

Second, in Figure 6C, expression of p53 in different tumors is examined. This shows a remarkable heterogeneity in outcomes with some tumors showing high levels of p53EE and others showing very little, even for the same tumor type. The authors should address these findings and discuss the basis for this and how it influences thinking about p53EE.

Third, Figure EV3 deals with a striking observation of increased senescence with p53EE cells. This is an underdeveloped aspect of the study that needs to be more fully addressed and discussed. There is especially a need to integrate this senescence finding with what is considered to be the molecular basis for p53EE functions.

Referee #2:

Comments for Authors: EMBOJ-2019-102096

Stiewe and colleagues present a comprehensive body of work and a series of elegant experiments that demonstrate that the R178E mutant of p53, which otherwise is totally defective in p53 target gene regulation, is stabilized and triggers mitochondrial-dependent apoptosis in the context of the Mdm2 deficiency. Further, this response is strikingly selective, as p53-R178E is insufficient to suppress spontaneous or oncogene-induced tumorigenesis; indeed the tumors in such mice can express very high levels of stabilized p53-R178E protein. Finally, the authors also show that p53-R178E tumors display sensitivity to chemotherapy, and show that human cells engineered to express the orthologous mutant p53 (p53-R181L), a recurrent mutant found in human patients, are similarly sensitive to chemotherapy. Given their findings the authors conclude that at least this class of mutant p53-driven tumors can retain apoptotic activity that can be exploited to robustly respond to chemotherapy.

The authors' findings are certainly provocative, and challenge the long-held dogma, and a very large body of literature, that mutant p53 confers resistance to therapy. Of course, as the authors note, the unique properties of the p53-R178E "cooperativity" mutant are likely distinct from others that disable the DNA binding domain (DBD), as such cooperativity mutants compromise p53 function by impairing formation of the salt bridge between two adjacent p53 subunits, rather than affecting DNA contacts or disrupting structure as seen in other hotspot mutants of p53. This is elegantly proven by the authors in their generation of double "p53-RR/EE" knock-in mice that also have a mutation in adjacent residue, E177R, that is capable of forming a salt bridge with p53-R178E to enable DNA binding and transcriptional activity. Furthermore, the findings of the authors are by no means clinically trivial, as the authors note that cooperativity mutants of p53 account for a staggering 27,000-34,000 cases per year (both numbers are given in the report). Accordingly, there is much enthusiasm for the unique knock-in p53-R178E mouse model and the studies the authors

have generated, which clearly establishes that DNA binding cooperativity is essential for both DNA binding and tumorigenesis.

The authors findings are convincing and provocative, and will certainly interest the readership of EMBO J. However, some aspects of this body of work need to be further developed before this study is suitable for publication.

Specific Suggestions:

1. The authors' provocative findings suggest that:

- Patients having p53 cooperativity mutants should have an improved overall or progression-free survival compared to structural or DNA binding contact p53 mutants. Is this the case?
- The selection for p53 mutations that are observed in many models and human patients following chemotherapy should exclude cooperativity mutants. Are there databases that allow the authors to address this important question?

2. Figure EV3C. While the effects of CRISPR deletion of p53EE on senescence bypass are evident in the data shown in Figure EV3E, CRISPR derivatives are not shown in Figure EV3C, as noted in the text on the bottom of page 8. This error should be rectified.

3. Figure 3 and Figure EV3. Direct demonstration of DNA damage should be shown to support the claims that elevated ROS is inducing DNA damage in the protracted senescence response of p53-EE/EE MEFs. This can be easily examined by gamma-H2AX foci by confocal immunofluorescence or flow cytometry, and by performing conventional comet assays.

4. The fact that p53-R178E might interact with and affect NRF2 functions in late passage p53EE MEFs and in the tumor models assessed is interesting and could be examined by evaluating the effects of p53-R178E on NRF2 targets genes and assessment of mitochondrial mass (e.g., using MitoTracker Green). This is important given that the authors' data imply non-transcriptional mechanisms involving direct mitochondrial priming are important for the observed phenotypes of p53-R178E.

5. Are there effects of p53-R178E on metabolism that might account for the ROS generation observed? At a minimum the authors should assess effects on metabolism (OCR and ECAR), for example using their elegant Tet-inducible system that allows one to conditionally express p53-R178E in the context of p53/Mdm2-double null cells.

Minor point:

Figures 2C and EV2A. The authors should document comparable expression of adenovirus E1A and Ha-Ras in p53^{+/+} vs. p53-EE/EE MEFs.

Referee #3:

Timofeev et al established a new p53 mutant mouse model with a point mutation at p53 DNA binding domain, which they named as R178E mutant. They showed that this mutant is defective in transcriptional activation, apoptosis, cell cycle arrest and senescence. However, this mutant still was capable of promoting apoptosis and embryonic lethality in Mdm2 deleted background, suggesting residual p53 activities of this mutant allele. While this mutant allele was unable to suppress tumor formation, it led to increased survival following chemotherapy. Together, these data suggest that there may be a subset of p53 mutations in cancer that would convey survival advantages in response to chemotherapy treatment.

Overall, this is a very nice manuscript with convincing data supporting their working hypothesis.

Potential survival benefit of some p53 mutations following chemotherapy is interesting and can be tested if the authors have access to clinical data with known p53 mutations.

1st Revision - authors' response

13th Jun 2019

Please see next page.

Rebuttal letter

We thank all three reviewers for their extremely helpful and constructive comments that allowed us to substantially revise and improve the manuscript. Please see below our answers to the individual reviewer comments.

Referee: 1

Previous studies from the authors' laboratory had identified specific residues on the p53 tumor suppressor that influence its ability to engage in cooperative sequence-specific DNA binding. In this manuscript, they utilize a new mouse model with a missense mutation within the endogenous p53 gene to provide three novel and interesting findings about p53 and its role in cancer. First, they clearly show that the ability of p53 to rescue the embryonic lethality of Mdm2 knockout mice is distinct from its role in tumor suppression. This supports the notion that the role of p53 in development is different from that in cancer. Second, they provide compelling evidence to support the idea that mutant p53-driven tumors will be responsive to chemotherapy in tissues and under conditions in which p53-null tumors are not. This has translational potential and high significance for designing personalized therapies using p53 status as a guide. Third, the data herein firmly establish a role for transcriptional-independent effects of p53 in tumor suppression. This idea has been examined extensively in the past but these authors now provide genetic evidence to separate cytoplasmic from nuclear roles for p53.

The manuscript is clearly written. Multiple and diverse approaches are used to support the findings. Given the significance and innovation, the study is quite suitable for The EMBO Journal. There are three concerns that should be addressed before publication.

First, a notable finding is the observation of embryonic lethality with Mdm2 knockout by the p53^{EE} mutant at E9.5. However, this is distinct from what is observed with p53 null embryos where lethality is observed at the blastocyst stage. The authors do not acknowledge or discuss this important difference. Data comparing the p53^{EE} mutant embryos to p53 null embryos is essential and needs to be included and discussed.

Trp53^{-/-};Mdm2^{-/-} embryos are not lethal. They are viable and largely normal, except for an increased tumor susceptibility. It is the Trp53^{+/+};Mdm2^{-/-} genotype that is lethal very early during development. While first reports mapped the lethality to shortly after implantation (Jones et al., 1995; and Montes de Oca Luna et al., 1995), later studies revealed a hatching defect and initiation of apoptosis already in E3.5 blastocysts prior to implantation (Chavez-Reyes et al., 2003). We have referenced this in our manuscript in lines 230-233.

What the reviewer might be alluding to is the marked time difference in the Mdm2-knockout lethality of p53^{EE} versus wild-type p53 embryos: Trp53^{EE/EE};Mdm2^{-/-} embryos die around E9.5, while Trp53^{+/+};Mdm2^{-/-} embryos die much earlier starting E3.5. This approximately one week delay in lethality of Trp53^{EE/EE};Mdm2^{-/-} embryos is in line with the severely reduced activity of the p53^{EE} mutant seen throughout our study. However, the fact that it is just a delay and – different from Trp53^{-/-};Mdm2^{-/-} embryos – not a complete rescue, points towards the residual cytotoxic activities of the p53^{EE} mutant that are interesting and unexpected. We have added this point to the results section.

Second, in Figure 6C, expression of p53 in different tumors is examined. This shows a remarkable heterogeneity in outcomes with some tumors showing high levels of p53^{EE} and others showing very little, even for the same tumor type. The authors should address this findings and discuss the basis for this and how it influences thinking about p53^{EE}.

Figure 6C was included to demonstrate the degree of heterogeneity of p53^{EE} expression observed in different tumors that have developed spontaneously in p53^{EE} mice. It should be noted, however, that the vast majority of tumors showed strong p53^{EE} staining. To more accurately demonstrate this, we have stained p53 in a panel of 36 tumors from p53^{EE/EE} mice and scored their p53 expression (new Fig. EV5). Only 17% showed no p53^{EE} staining, whereas 67% showed a high staining score of 2-3. This is comparable to the p53 staining described for spontaneous tumors arising in R172H-mutant mice (Terzian et al., 2008).

Although positive p53 immunostaining is frequently considered diagnostic of a p53 mutations in human cancer patients, it is well known that not all p53-mutated tumors are immuno-positive for p53. Our finding that the majority, but not all, p53EE tumors show strong p53EE staining therefore not only resembles what is seen in mouse models for p53 hotspot mutants but also reflects what is typically seen in clinical samples.

Of note, E μ -Myc lymphomas and AML1-ETO/NrasG12D-driven leukemias were all positive for p53EE expression in immunohistochemistry. As mouse studies on p53 hotspot mutants have mechanistically linked mutant p53 stabilization to Mdm2 inhibition and oncogenic signaling (Terzian et al., 2008), we assume that the enforced expression of strong oncogenes such as Myc and Nras invariably causes p53EE stabilization in these models. In contrast, levels of oncogenic signaling in spontaneously arising tumors are likely more variable and translate into a more heterogenous p53EE expression. We have added a sentence on this to the results section.

Third, Figure EV3 deals with a striking observation of increased senescence with p53EE cells. This is an underdeveloped aspect of the study that needs to be more fully addressed and discussed. There is especially a need to integrate this senescence finding with what is considered to be the molecular basis for p53EE functions.

We agree that the increased senescence observed in late-passage p53EE MEFs is mechanistically still poorly understood. For reasons of completeness, we nevertheless decided to include this information as "extended data" being aware that it is mechanistically not completely worked out. As requested also by referee 2, we have now included some more experimental data that provide more insight into the underlying mechanisms.

Given that our data indicated a role for oxygen and ROS in the late-passage senescence phenotype, we evaluated the possibility that p53EE either promotes an oxidative metabolism or limits the cellular anti-oxidative defense. We first performed the metabolism analysis requested by referee 2 and measured the effect of p53EE on the oxygen consumption rate (OCR) of Trp53^{-/-};Mdm2^{-/-} MEFs. This analysis showed a statistically non-significant trend towards a more glycolytic, but not oxidative metabolism. Importantly, the analysis showed a significantly reduced spare respiratory capacity pointing towards inhibitory effects on mitochondrial function. We have included this result as Fig. EV3I.

As also requested by referee 3, we measured mitochondrial mass using qPCR for mitochondrial genes as a surrogate marker (Fig. EV3J). In line with the observed reduction in spare respiratory capacity, mitochondrial DNA/mass was significantly reduced in p53EE MEFs especially at later passages.

As other p53 mutants were shown to block Nrf2 function critical for ROS defense but also mitochondrial respiratory function and mitochondrial biogenesis, we proceeded to evaluate Nrf2-function comparatively in p53^{-/-} and p53EE MEFs. First, we noted that late-passage p53EE MEFs, despite having elevated ROS (Fig. EV3F), did not show Nrf2-mediated upregulation of ROS-induced genes (data not shown). In fact, p53EE MEFs showed lower basal expression of multiple Nrf2 target genes (Fig. EV3K) and, compared to p53^{-/-} MEFs, strongly impaired induction of the highly ROS-responsive Nrf2-target gene *Hmox1* when exposed to hydrogen peroxide (Fig. EV3L).

Together these new results confirm our previous speculations that p53EE interferes with the Nrf2-mediated anti-oxidative response and provide an explanation why p53EE MEFs undergo late-passage senescence in an oxygen-dependent manner. We have added these new data to Fig. EV3 and the resulting implications to our discussion.

Referee: 2

Stiewe and colleagues present a comprehensive body of work and a series of elegant experiments that demonstrate that the R178E mutant of p53, which otherwise is totally defective in p53 target gene regulation, is stabilized and triggers mitochondrial dependent apoptosis in the context of the Mdm2 deficiency. Further, this response is strikingly selective, as p53-R178E is insufficient to suppress spontaneous or oncogene-induced tumorigenesis; indeed the tumors in such mice can express very high levels of stabilized p53-R178E protein. Finally, the authors also show that p53-R178E tumors display sensitivity to chemotherapy, and

show that human cells engineered to express the orthologous mutant p53 (p53-R181L), a recurrent mutant found in human patients, are similarly sensitive to chemotherapy. Given their findings the authors conclude that at least this class of mutant p53-driven tumors can retain apoptotic activity that can be exploited to robustly respond to chemotherapy. The authors' findings are certainly provocative, and challenge the long-held dogma, and a very large body of literature, that mutant p53 confers resistance to therapy. Of course, as the authors note, the unique properties of the p53-R178E "cooperativity" mutant are likely distinct from others that disable the DNA binding domain (DBD), as such cooperativity mutants compromise p53 function by impairing formation of the salt bridge between two adjacent p53 subunits, rather than affecting DNA contacts or function by impairing formation of the salt bridge between two adjacent p53 subunits, rather than affecting DNA contacts or disrupting structure as seen in other hotspot mutants of p53. This is elegantly proven by the authors in their generation of double "p53-RR/EE" knock-in mice that also have a mutation in adjacent residue, E177R, that is capable of forming a salt bridge with p53-R178E to enable DNA binding and transcriptional activity. Furthermore, the findings of the authors are by no means clinically trivial, as the authors note that cooperativity mutants of p53 account for a staggering 27,000-34,000 cases per year (both numbers are given in the report). Accordingly, there is much enthusiasm for the unique knock-in p53-R178E mouse model and the studies the authors have generated, which clearly establishes that DNA binding cooperativity is essential for both DNA binding and tumorigenesis.

The authors findings are convincing and provocative, and will certainly interest the readership of EMBO J. However, some aspects of this body of work need to be further developed before this study is suitable for publication.

Specific Suggestions:

1. The authors' provocative findings suggest that:

- Patients having p53 cooperativity mutants should have an improved overall or progression-free survival compared to structural or DNA binding contact p53 mutants. Is this the case?
- The selection for p53 mutations that are observed in many models and human patients following chemotherapy should exclude cooperativity mutants. Are there databases that allow the authors to address this important question?

We absolutely agree with the referee's suggestions on the clinical implications of our findings and have thoroughly explored the possibilities to test these predictions. However, the overall low frequency of cooperativity mutations precludes this type of analysis.

To illustrate the problem: the yearly 27,000 cases with a R181 mutation or 34,000 cases with a cooperativity mutation (either E180 or R181) are estimates calculated based on the world-wide cancer incidence published in the IARC World Cancer Report 2014 and a frequency of approximately 0.5% for cooperativity mutations among all p53 mutations. The UMD TP53 mutation database lists a total of 80,406 p53-mutated cases, 417 of which are affecting codons 180/181 (0.52%). The IARC p53 database lists a total of 27,849 cases, 160 of which are codon 180/181 mutations (0.57%). For comparison, the most frequent R175H mutation has a frequency of 4.3% in the UMD TP53 database. To avoid any confusion, we have revised the manuscript and only present the estimate of 34,000 cases for cooperativity mutations affecting codons 180/181.

Nevertheless, this frequency is not sufficient for survival analysis, even when analyzing the complete TCGA pan-cancer cohort comprising 41,738 non-redundant samples of which 12,899 (35%) are p53-mutated. Similar to the 0.5% of cooperativity mutations in the UMD and IARC p53 mutations databases, 53 TCGA cases (0.41% of all p53 mutations) are codon 180/181 mutations. However, survival data are only available for 3 of these 53 cooperativity-mutant cases. It is obvious that even the large TCGA dataset is insufficient to correlate cooperativity mutations with survival or therapy response and draw meaningful conclusions regarding their clinical impact.

2. Figure EV3C. While the effects of CRISPR deletion of p53EE on senescence bypass are evident in the data shown in Figure EV3E, CRISPR derivatives are not shown in Figure EV3C, as noted in the text on the bottom of page 8. This error should be rectified.

We have rephrased the sentence accordingly. Figure EV3C was referenced only for the spontaneous p53EE deletion.

3. Figure 3 and Figure EV3. Direct demonstration of DNA damage should be shown to support the claims that elevated ROS is inducing DNA damage in the protracted senescence response of p53-EE/EE MEFs. This can be easily examined by gamma-H2AX foci by confocal immunofluorescence or flow cytometry, and by performing conventional comet assays.

Unfortunately, we did not have aged p53EE MEFs in culture and passaging fresh MEFs until reaching senescence would have extended beyond the time frame available for the revision. We therefore cannot formally demonstrate ROS-induced DNA damage and have removed the statements regarding DNA damage.

4. The fact that p53-R178E might interact with and affect NRF2 functions in late passage p53EE MEFs and in the tumor models assessed is interesting and could be examined by evaluating the effects of p53-R178E on NRF2 target genes and assessment of mitochondrial mass (e.g., using MitoTracker Green). This is important given that the authors' data imply non-transcriptional mechanisms involving direct mitochondrial priming are important for the observed phenotypes of p53-R178E.

To directly address whether Nrf2 function is compromised by p53EE we have measured expression of Nrf2 target genes in fresh p53^{-/-} and p53EE MEFs. We show that basal expression of multiple canonical anti-oxidative Nrf2 target genes is substantially down-regulated in p53EE MEFs and not properly activated when cells are exposed to ROS, i.e. treated with hydrogen peroxide (Fig EV3K,L). Furthermore, we measured mitochondrial DNA content using quantitative PCR for mitochondrial genes as a surrogate marker for mitochondrial mass, as we had genomic DNA from the early and late passage MEF cultures that were used for the other experiments depicted in Fig EV3. p53EE MEFs showed a significant reduction of mitochondrial DNA content which became even more evident at late passages (Fig. EV3J). As Nrf2 is critical not only for ROS defense but also for mitochondrial respiratory function, biogenesis and integrity, the reduced mitochondrial DNA/mass is in line with Nrf2-inhibition by p53EE. We therefore agree with the referee's interpretation that Nrf2 inhibition might not only explain p53EE effects in senescence but also be implicated in mitochondrial priming for apoptosis and have added this point to our discussion.

5. Are there effects of p53-R178E on metabolism that might account for the ROS generation observed? At a minimum the authors should assess effects on metabolism (OCR and ECAR), for example using their elegant Tet-inducible system that allows one to conditionally express p53-R178E in the context of p53/Mdm2-double null cells.

We have done the requested experiment (Fig EV3I). Although p53WT is known to promote oxidative phosphorylation and counteract the Warburg effect, there is no evidence for a metabolic shift towards oxidative phosphorylation upon conditional expression of p53EE in the context of p53/Mdm2-double null cells. In fact, there is a (statistically non-significant) trend in the opposite direction: reduced basal oxygen consumption and oxidative ATP production upon p53EE expression. We therefore believe, that the increase in ROS in p53EE MEFs is not a consequence of residual p53WT-like stimulation of oxidative phosphorylation. However, we noted a statistically significant decrease in the spare respiratory capacity that is consistent with inhibitory p53EE effects on mitochondrial function and Nrf2 activity as described above regarding comment 4.

Minor point:

Figures 2C and EV2A. The authors should document comparable expression of adenovirus E1A and Ha-Ras in p53^{+/+} vs. p53-EE/EE MEFs.

Western Blots demonstrating comparable E1A and Ras expression have been included.

Referee: 3

Timofeev et al established a new p53 mutant mouse model with a point mutation at p53 DNA binding domain, which they named as R178E mutant. They showed that this mutant is defective in transcriptional activation, apoptosis, cell cycle arrest and senescence. However, this mutant still was capable of promoting apoptosis and embryonic lethality in Mdm2 deleted background,

suggesting residual p53 activities of this mutant allele. While this mutant allele was unable to suppress tumor formation, it led to increased survival following chemotherapy. Together, these data suggest that there may be a subset of p53 mutations in cancer that would convey survival advantages in response to chemotherapy treatment.

Overall, this is a very nice manuscript with convincing data supporting their working hypothesis. Potential survival benefit of some p53 mutations following chemotherapy is interesting and can be tested if the authors have access to clinical data with known p53 mutations.

As much as we would have liked to test the clinical implications on survival following chemotherapy, there are just not enough survival data available to draw any statistically sound conclusions. Please also see our more detailed response to comment 1 from referee #2.

2nd Editorial Decision

25th Jul 2019

Thank you for submitting your revised manuscript for consideration by The EMBO Journal. Please accept my apologies for the delay in processing your revised manuscript due to protracted referee input. Your revised study was sent back to the referee #2 for re-evaluation, and we have received his-her comments, which I enclose below. As you will see the referee finds that the concerns have been sufficiently addressed and is are now broadly in favour of publication.

Thus, we are pleased to inform you that your manuscript has been accepted in principle for publication in The EMBO Journal, pending some minor issues related to formatting and data representation as listed below, which need to be adjusted at re-submission.

REFeree REPORTS:

Referee #2:

Comments for Authors: EMBOJ-2019-102096R

The revised and comprehensive study of Thorsten Stiewe and colleagues is viewed as a very important body of work that significantly advances our understanding of the "cooperativity subgroup" of mutant p53 that retain wild-type p53 apoptotic activity but that are defective in other p53 tumor suppressor functions. In addition to the impressive studies previously provided, the authors have now added new and important studies that indicate that p53-R178E interactions with Nrf2 impair Nrf2-mediated anti-oxidant responses and functions (e.g., reductions in Nrf2 target genes and mitochondrial mass), which likely account for the elevated ROS, senescence and mitochondrial priming phenotypes provoked by such p53 mutants. The authors have also appropriately addressed all of the concerns of the Reviewers. Finally, this study is certain to interest the readership of the EMBO Journal.

2nd Revision - authors' response

30th Jul 2019

The authors performed the requested editorial changes.

Corresponding Author Name: Thorsten Stiewe

Manuscript Number: 2019-102096